# Modelling the marine ecosystem of IBI European waters for CMEMS operational applications

Elodie Gutknecht[1], Guillaume Reffray[1], Alexandre Mignot[1], Tomasz Dabrowski[2], and Marcos Garcia-Sotillo[3]

[1] Mercator Ocean, Parc Technologique du Canal, 8-10 rue Hermes, 31520 Ramonville St-Agne, France
[2] Marine Institute, Rinville, Oranmore, Co. Galway, H91 R673, Ireland
[3] Puertos del Estado, Av. Partenón, 10, 28042 Madrid, Spain

*Correspondence to*: Elodie Gutknecht (elodie.gutknecht@mercator-ocean.fr)

**Abstract.** As part of the Copernicus Marine Environment Monitoring Service (CMEMS), a physical-biogeochemical coupled model system has been developed to monitor and forecast the ocean dynamics and marine ecosystem of the European waters, and more specifically on the Iberia-Biscay-Ireland (IBI) area. The CMEMS IBI coupled model covers the North-East Atlantic Ocean from the Canary Islands to Iceland, including the North Sea and the Western Mediterranean, with a NEMO-PISCES 1/36° model application. The coupled system has provided 7-day weekly ocean forecasts for CMEMS since April 2018. Prior to its operational launch, a pre-operational qualification simulation (2010-2016) has allowed assessing the model's capacity to reproduce the main biogeochemical and ecosystem features of the IBI area. The objective of this paper is then to describe the consistency and skill assessment of the PISCES biogeochemical model, using this 7-year qualification simulation. The model results are compared with available satellite estimates as well as *in situ* observations (ICES, EMODnet and BGC-Argo).

The simulation successfully reproduces the spatial distribution and seasonal cycles of oxygen, nutrients, chlorophyll-a and net primary production; and confirms that PISCES is suitable at such a resolution and can be used for operational analysis and forecast applications. This model system can be a useful tool to better understand the current state and changes in the marine biogeochemistry of European waters and can also provide key variables for developing indicators to monitor the health of marine ecosystems. These indicators may be of interest to scientists, policy makers, environmental agencies and the general public.

## 1 Introduction

The North-East Atlantic waters are subject to natural climate variability as well as intense human pressures that can have significant impacts for the marine ecosystem. In addition to intense fishing activity, human pressures also include aquaculture, agriculture, maritime transport, oil and gas extraction, tourism and urbanization. In order to regulate, sustainably manage, protect and conserve the maritime areas of the North-East Atlantic waters, the European Union adopted the OSPAR Convention in 1992. The European Union has also set up an Earth observation programme, the Copernicus European Programme, formerly known as GMES (Global Monitoring for Environment and Security). Copernicus aims for developing an operational and autonomous Earth observation capacity of European Union to serve the general interest and help public authorities and other international organizations to improve the quality of life. Its marine monitoring component, the Copernicus Marine Environment Monitoring Service (CMEMS; http://www.marine.copernicus.eu), is coordinated and led by Mercator Ocean, a service provider of ocean information in real and delayed time (http://www.mercator-ocean.eu). Gathering satellite*, in situ* and model data, CMEMS provides regular and systematic information on the state and variability of the ocean dynamics and marine ecosystems for the global ocean and the European regional seas over six different areas: Arctic Ocean, Baltic Sea, European North-West Shelf Seas, Black Sea, Mediterranean Sea and Iberia–Biscay–Ireland (IBI) Seas.

The CMEMS IBI Monitoring and Forecasting Center (IBI-MFC) is in charge of delivering multi-decadal reanalysis and operational analysis and short-term forecasts over the IBI European waters. It is managed by a consortium of Centres, coordinated by Puertos del Estado and that includes Mercator Ocean, Meteo France, the Spanish Met office AEMET, the Irish Marine Institute and the CESGA Supercomputing Centre. The IBI area covers part of the North-East Atlantic Ocean from the Canary Islands to Iceland, the North Sea and the Western Mediterranean (hereinafter referred as IBI Extended Domain). However the IBI-MFC delivers IBI products to CMEMS end-users over a smaller area, extending from -19°E to 5°E and 26°N to 56°N (hereinafter referred to as IBI Service Domain). IBI Extended and Service Domains are shown in Fig. 1a, and further details on the IBI-MFC and the IBI region definition are available in Sotillo et al. (2015).

To reach the IBI-MFC needs, Mercator Ocean has developed an analysis and forecast system adapted to the IBI area, which is a complex region to simulate for numerical models because the highly variable bathymetry gives rise to a wide spectrum of physical and biogeochemical ocean processes. The IBI ocean dynamics and thermodynamics are resolved by the 3.6 version of the NEMO modelling platform (Madec et al., 1998; Madec, 2008) at 1/36° spatial resolution and constrained through data assimilation of *in situ* and satellite physical data. The oceanic passive tracers of the lower trophic levels ecosystem dynamics is simulated by the PISCES biogeochemical model (Aumont et al., 2015). NEMO and PISCES are "online" coupled on the same 1/36° model grid resolution, placing the model into the sub-mesoscale-permitting regime. This resulting system, hereinafter referred to as IBI36, has been used to produce a pre-operational qualification simulation that has served as initial conditions to start the operational analysis and forecast system. This system has provided on a weekly basis a short-term (7-days) forecast of the ocean dynamics and the main biogeochemical variables of the marine ecosystem, since April 2018. The physical and biogeochemical products are disseminated through the CMEMS web site, and described and validated in Maraldi et al. (2013), Sotillo et al. (2015, 2018), Bowyer et al. (2018), and Amo et al. (2018, 2019).

PISCES simulates the lower trophic levels of the marine food web, from nutrients to mesozooplankton. It has already been successfully used in various biogeochemical studies at global, regional scales and up to process studies, at low and high spatial resolutions as well as for short-term and long-term analyses (e.g. Bopp et al., 2005; Gehlen et al., 2006, 2007; Schneider et al., 2008; Steinacher et al., 2010; Tagliabue et al., 2010, Séférian et al., 2013; Gutknecht et al., 2016). PISCES is also used in operational oceanography (Brasseur et al., 2009), for the CMEMS global ocean analysis and forecast system at ¼° (Perruche et al., 2016) and the Indonesian seas operational system at 1/12° (the INDESO project; Gutknecht et al., 2016).

Although PISCES has been used so far to answer a wide range of scientific questions, it has never been used at such a resolution before. First of all, due to its high resolution, the IBI36 system represents a challenge in terms of numerical, computational and operational constraints. Moreover, although the biogeochemical equations of PISCES remain unchanged between the CMEMS global ocean system at ¼° and IBI36, the distribution of biogeochemical tracers is impacted by a better resolution of ocean dynamics, notably through finer bathymetry and a more accurate coastline. In addition, IBI36 considers small-scale phenomena and high-frequency processes of primary importance such as tides and atmospheric pressure forcing. The detailed validation of marine processes at the regional and coastal scale by Maraldi et al. (2013) showed the model's ability to correctly reproduce tidal fronts, narrow boundary currents, surges… All these contrasting dynamic regimes create different biogeochemical environments that impact the availability of nutrients, oxygen and light for phytoplankton growth, thus impacting the rest of the marine food chain.

In addition to IBI36, European waters are also covered by other biogeochemical models, as several MFC share a part of their model domain with IBI. The operational system for the Northwest European Shelf Seas (Edwards et al., 2012; O'Dea et al., 2017) is based on NEMO and the ERSEM biogeochemical component (Blackford et al., 2004; Butenschönet al., 2016) at 1/15° latitudinal resolution and 1/9° longitudinal resolution (~ 7km). The Mediterranean Sea forecasting system (Lazzari et al., 2012; Tonani et al., 2014; Teruzzi et al., 2018; Salon et al., 2019) is built on NEMO and the BFM biogeochemical model (Vichi et al., 2013) at 1/24° resolution. These regional coupled systems, and of course the 1/4° NEMO-PISCES global ocean

operational system, are the subject of inter-comparisons. Ocean dynamics forecast models, all of them based on NEMO, are already being compared in Lorente et al. (2019) and Mason et al. (2019). For the marine biogeochemistry and ecosystem dynamics, inter-comparison is initiated in the framework of CMEMS, but represents a significant work because the biogeochemical models differ in complexity (Gehlen at al., 2015).

In this paper, the pre-operational qualification simulation (which extends from January 2010 to December 2016) of the IBI36 system is evaluated using different observational data sets (satellite, oceanographic historical databases and BGC-Argo float network). This paper represents the first validation of the biogeochemical component of the IBI36 system. The objective is to assess the performance of the PISCES model in reproducing the main biogeochemical characteristics of IBI European waters, and verify that PISCES is suitable at such a resolution and can be used for operational analysis and forecast applications. Oxygen, nutrients, chlorophyll-a (hereinafter denoted Chl-a) and net primary production (NPP) are assessed in terms of consistency and quality (or accuracy) using GODAE-like metrics (Hernandez et al., 2009, 2015). Despite some of the areas presented here are outside of the IBI Service Domain (where the IBI-MFC delivers IBI products to CMEMS end-users), the model is evaluated over the IBI Extended Domain in order to take advantage of the *in situ* observation coverage.

The paper is organized as follows. Section 2 presents an overview of the IBI European seas with emphasis on the main drivers of the ecosystem dynamics. Section 3 describes the IBI36 system, the model initialization and boundary conditions, the external forcing, the regional adaptations and the data used for evaluation. In Sect. 4, the biogeochemical tracers are compared to satellite and *in situ* observations. Finally, Discussion and Conclusions are provided, respectively, in Sect. 5 and 6.

## 2 IBI European waters

Phytoplankton dynamics is controlled by the complex interaction between ocean dynamics, nutrients and light availability. The biogeochemical and ecosystem dynamics of IBI European waters are synthesized in this section, with an emphasis on several areas (please see the 12 black boxes added to Fig. 1a). These areas are adopted in the validation framework (Section 4.1), and are named throughout the following description.

In the IBI European waters, phytoplankton dynamics usually follows a seasonal cycle typical of temperate seas, governed by the alternation between winter mixing and summer stratification of the water column (Barton et al., 2015). A rapid increase in phytoplankton biomass starts in spring, when seasonal re-stratification begins and when the Mixed Layer Depth (MLD) becomes shallower (Sverdrup 1953; Behrenfeld 2010; Taylor and Ferrari 2011). This spring bloom is followed by a summer decrease in biomass, when the increase in stratification of the water column reduces the vertical supply of nutrient to the euphotic layer (the layer where phytoplankton grows) (Barton et al., 2015).

In the North-East Atlantic Ocean, primary productivity increases from south to north (boxes 3, 2 and 1 in Fig. 1a). In the subtropical North Atlantic, wind stress induces Ekman downwelling that deepens the nutricline and the warm waters maintain a stratification of the water column throughout the year (Barton et al., 2015). The annual primary production is then limited and so is its seasonal variations. The subtropical gyre is separated from the subpolar gyre by the meandering Subpolar Front which covers a relatively wide region (Rossby, 1996), and represents a transition zone between the two regimes. In the subpolar North Atlantic, the seasonal surface cooling deepens the mixed layer in winter, winds drive Ekman upwelling and make the nutricline shallower (Barton et al., 2015). However, light supply limits the phytoplankton growth in winter. A strong spring bloom is triggered by water column re-stratification in spring, while during summer the stratification limits the nutrient supply to the surface (Williams et al. 2000). This seasonal upward flow of deep and nutrient-rich waters triggers a higher productivity and a strong seasonal cycle.

Moving toward the coast, Moroccan and Iberian upwelling systems (boxes 10 and 8 in Fig. 1a) are part of the Canary Current Upwelling System, one of the four main Eastern Boundary Upwelling Systems of the world, and thus a very

productive ecosystem and an active fishery (Aristegui et al., 2004). The season for upwelling along the Iberian coast begins in May-June with the establishment of northerly winds and continues throughout the summer (June-September; Wooster et al., 1976; Nykjær and Van Camp, 1994). Along the Moroccan coasts, upwelling intensifies from the north, where it is highly seasonal, to the south where it can be considered permanent and intense, with maximum activity from April to September (Pelegrí and Benazzouz, 2015).

The IBI European waters also cover part of the Western Mediterranean Sea (boxes 11 and 12 in Fig. 1a). From a biogeochemical perspective, the Mediterranean Sea shows a high N:P ratio (N:P ~ 20 for the western basin; Ribera d'Alcalà et al., 2003; Lazzari et al., 2016) and relatively high oxygen consumption rates compared to the Atlantic and Pacific Oceans (Christensen et al., 1989; Roether and Well, 2001). Mediterranean Outflow Water (MOW) flows into the Gulf of Cadiz (box 9 in Fig. 1a) and the Atlantic through a sill depth of only 290 meters at the Strait of Gibraltar. This salty and denser water
flows out at the bottom of the sill and a northward-moving MOW core spreads on the continental slope of Portugal at 1000 meters depth, enters the Bay of Biscay and follow the shelf break further north.

The Bay of Biscay and Celtic Seas are moderately productive ecosystems (UNEP LME report, 2008). The seasonal cycle of phytoplankton in the Bay of Biscay (box 7 in Fig. 1a) is typical of temperate seas (Fernández and Bode, 1991; Valdés et al., 1991; Lavín et al., 2006) but spatial variability is high. The bay is characterized by a weak anticyclonic circulation in the
oceanic part, a coastal upwelling, the northerly flow of MOW (OSPAR, 2000; Lavín et al., 2006) and river discharges (Gohin et al., 2003). In the oceanic part of the bay, a major biomass peak can be observed in spring due to oligotrophic conditions. However, in the coastal part of the bay, phytoplankton remains relatively high during winter for isobaths less than 100 meters in the Region Of Freshwater Influences (ROFI).

The continental shelf widens in the Bay of Biscay. It is quite narrow along the Spanish coast, but increases rapidly with
latitude along the French coasts, from 10 km in the south to more than 200 km wide in the north of the bay. The most extensive continental shelf areas are in the Celtic Seas and the North Sea. The continental shelf along the European coasts is the site of intense tidal amplitude and turbulent mixing that prevent stratification (Piraud et al., 2003; Lam et al., 2003, Lavín et al., 2006). To the west of the Celtic Seas, a significant and permanent front can be observed in Chl-a at the edge of the shelf, extending from the northern Bay of Biscay to the Faroe-Shetland Channel and associated with the shelf edge current
(Belkin et al., 2009; Aquarone et al., 2008). The English Channel (box 6 in Fig. 1a), connecting the North Sea to the Atlantic, is generally mixed and strongly influenced by winds. The North Sea (boxes 4 and 5 in Fig. 1a) is characterized by significant river discharges and permanently mixed water column in the south, supplying the highest coastal primary production rates. The north part is characterized by a seasonal stratification and a deep channel in the north-east. Finally, as eastern boundary of the IBI domain, the Skagerrak and Kattegat connect the North Sea to the Baltic Sea.

Discharges of fresh and nutrient-rich water from rivers are a strong forcing factor for European waters. In addition to natural inputs related to the watershed erosion, many European coastal ecosystems are damaged by eutrophication due to human activities such as wastewater, agriculture and fish farming (Valdés and Lavin 2002). Eutrophication affects coastal areas, fjords and estuaries, mainly within the Celtic Seas, the Bay of Biscay and the Iberian Coast (OSPAR, 2003). Excessive nutrient enrichment, usually due to increased nitrogen and phosphorous concentrations in rivers, leads to high primary
production rates and reduced oxygen concentrations in the bottom water. Oxygen deficiency was reported in the bottom waters of the North-West European shelf (OSPAR, 2013; Ciavatta et al., 2016) and can be used as indicator of the health of marine ecosystems.

## 3 The IBI36 configuration

### 3.1 The coupled model system

Within the framework of CMEMS, the IBI-MFC Team has deployed an operational forecast service based on a coupled physical-biogeochemical model application. The model domain covers part of the North-East Atlantic Ocean from the Canary Islands (26°N) to Iceland (64°N) and from 20°W to North Sea (14°E) and the Western Mediterranean (10°E), using a curvilinear grid (Fig. 1a) with a horizontal resolution of 1/36°, corresponding to ~2 km for latitudes covered by the IBI domain, and 50 vertical levels.

As already discussed in the Introduction, the physical model is based on the NEMO 3.6 hydrodynamic model (Madec et al., 1998; Madec, 2008), developed by the NEMO consortium. The NEMO modelling system is freely available (http://www.nemo-ocean.eu) and specific numerical choices include time-splitting and non-linear free surface to correctly simulate high frequency processes such as tides. The ocean dynamics is constrained through data assimilation of *in situ* and satellite physical data (temperature and salinity vertical profiles, sea surface height and sea surface temperature). The IBI36

physical component is described in Maraldi et al. (2013), Sotillo et al. (2015, 2018) and Amo et al. (2018); and the data assimilation method is described in Amo et al. (2018).

The biogeochemical model PISCES v2 (Aumont et al., 2015), part of the NEMO 3.6 modelling platform, is an intermediary complexity model taking into account 24 prognostic variables. The model considers five nutrients that limit phytoplankton growth (nitrate, ammonium, phosphate, silicate and iron) and four living compartments: two phytoplankton size classes

(nanophytoplankton and diatoms) and two zooplankton size classes (microzooplankton and mesozooplankton); the bacterial pool is not explicitly modelled. PISCES distinguishes three non-living detrital pools for organic carbon, particles of calcium carbonate and biogenic silicate. In addition, the model simulates the carbonate system and dissolved oxygen. Biogeochemical parameters are based on the standard parameters of PISCES v2. Please refer to Aumont et al. (2015) for the full description of the model.

For this regional configuration, physics and biogeochemistry are running simultaneously ("on-line" coupling), with the same 1/36° spatial resolution. For reason of numerical cost (optimization of the computing time), the numerical scheme for biogeochemical processes is forward in time (Euler) while the physical component uses the leap-frog scheme. To respect the mass conservation, the coupling between biogeochemical and physical components is done every other time. The time step of the biogeochemical model is therefore twice that of the physical component, i.e. 300 s. The advection scheme for

biogeochemistry is the same QUICKEST scheme (Leonard, 1979) used for the physical part, but using the limiter of Zalezak (1979). The IBI36 biogeochemical component is described in Bowyer et al. (2018) and Amo et al. (2019).

### 3.2 Model initialization, external forcing and boundary conditions.

The pre-operational qualification simulation starts on January 6, 2010 and runs until December 31, 2016. Ocean dynamics (temperature, salinity, currents and free surface) is initialized and forced to the open boundaries by the daily outputs of the

NEMO global ocean analysis and forecasting system at 1/12° (Lellouche et al., 2016, 2018) of CMEMS. Both regional and global systems are forced every 3 hours with atmospheric fields from the ECMWF. The biogeochemistry is initialized with the NEMO-PISCES global ocean analysis and forecasting system at ¼° (Perruche et al., 2016) of CMEMS for the same date, and open boundary conditions come from this same global product on a weekly basis. The global biogeochemical system is also forced by the coarsened solution of the global physics system mentioned just above, making the global and IBI

components of CMEMS consistent.

Other boundary fluxes account for the external supply of nutrients and carbon from three different sources. The model includes the atmospheric dust deposition of Fe, Si, P and N at the ocean surface (Aumont et al., 2015). River discharges of nutrients come from the Global NEWS 2 data sets (Mayorga et al., 2010) and carbon comes from Ludwig et al. (1996). An

iron source corresponding to sediment reductive mobilisation on continental margins is also considered. For more details on external nutrient supplies, please refer to Aumont et al. (2015).

Two adaptations are necessary in order to meet regional specificities. The first adaptation concerns the becoming of particles reaching the bottom boundary. Within the standard version of PISCES v2, the exchange between the ocean and the sediments assumes that a fraction of particulate material reaching the sea floor is permanently buried in the sediments while the remaining organic matter is dissolved or degraded and released into the water column. Concerning the IBI configuration, strong tidal currents prevent organic matter from settling on the bottom and being stored in the sediments over much of the North-West European continental shelf (De Haas et al., 2002). Thus, no permanent burial to the sediments is considered in the IBI36 system. The second adaptation concerns the supply of nutrients from rivers. As mentioned above, nutrient inputs come from the annual climatology at ½° spatial resolution of Global NEWS 2. They represent a realistic hydrology for the reference year 2000, considered as representative of the contemporary conditions (Mayorga et al., 2010). Inputs are injected into the model in the form of surface runoff in the river plumes of the Rhone River and the German Bight and along the coastline for other rivers, with caution of conserving the nutrient flows estimated by Global NEWS 2. However Global NEWS 2 seems to underestimate nutrient runoffs in the Western Europe (Mayorga et al., 2010). The only contribution of Global NEWS 2 is not sufficient to support the high coastal biological production of the IBI European waters (not shown). Additional inputs of nitrate and phosphate are then introduced into the system at source points of the 33 main rivers of the IBI Extended Domain (please refer to Maraldi et al. (2013) for the location of the rivers), and are linked to the physical flow. These additional nutrients come from rivers monitored and listed by the European Environment Agency (www.eea.europa.eu) on the basis of annual averages. This adaptation leads to higher coastal Chl-a and allows the model to reproduce the maximum Chl-a observed along the European coasts (not shown). It also allows representing the nutrients in excess likely to cause eutrophication in downstream coastal waters and oxygen deficiency in the bottom waters.

**3.3 Satellite and *in situ* observational data sources used for model validation**

The model results are compared with satellite and *in situ* observational data. Chl-a and NPP are derived from remote sensing estimations. Dissolved oxygen, nutrients (nitrate, phosphate, silicate and ammonium) and Chl-a concentrations are gathered in regional databases such as ICES (International Council for the Exploration of the Sea), EMODnet (European Marine Observation and Data Network), and the Biogeochemical-Argo (BGC-Argo) floats. Chl-a concentration is expressed in in mg Chl $m^{-3}$. NPP is expressed in mg C $m^{-2}$ $d^{-1}$. Oxygen and nutrient concentrations, for standardization purposes, are converted in µmol $l^{-1}$. The spatial distribution of ICES, EMODnet and BGC-Argo data are presented in Fig. 1b.

Remote sensing estimations of surface Chl-a are provided by the Ocean Colour - Climate Change Initiative project of the European Space Agency (ESA OC-CCI product), distributed via CMEMS. The regional ESA OC-CCI product for the North Atlantic Ocean has a resolution of 1 km. It merges SeaWiFS, MODIS-Aqua, MERIS and VIIRS sensors. Chl-a (in mg Chl $m^{-3}$) is estimated from the OC5CI regional algorithm case1/case2, a combination of OCI (Hu et al., 2012) and OC5 (Gohin et al., 2008). A combined algorithm is required because wide and shallow North-West European shelf seas are supplied in sediment and organic material by many estuaries, which makes the water turbid and disturbs the measurement of Chl-a concentrations. A detailed description of the ESA OC-CCI processing system can be found in Sathyendranath et al. (2012).

Three NPP products using the ocean colour data of the MODIS ocean colour sensor are distributed by the Oregon State University (www.science.oregonstate.edu/ocean.productivity): the Vertically Generalized Production Model (VGPM; Behrenfeld and Falkowski, 1997; usually recognized as the Standard product), an "Eppley" version of the VGPM product (Eppley-VGPM; Behrenfeld and Falkowski, 1997) and the Carbon-based Production Model (CbPM; Westberry et al. 2008). These global ocean estimates are monthly averages with a resolution of 1/6° and are expressed in mg C $m^{-2}$ $d^{-1}$. Due to the high uncertainty in NPP products (Henson et al., 2010; Emerson, 2014), PISCES estimates are compared with the three products mentioned above.

The ICES oceanographic database ([www.ices.dk/marine-data/data-portals](www.ices.dk/marine-data/data-portals)) gathers quality-controlled *in situ* observational data for the North-East Atlantic Ocean, the North Sea, the Baltic Sea and the Arctic Ocean. Dissolved oxygen, nitrate, phosphate, silicate, ammonium are all expressed in µmol $l^{-1}$ and Chl-a in mg Chl $m^{-3}$. Over the period of the IBI36 pre-operational qualification simulation, ICES data are mainly located in the shallow and coastal waters of the Northern seas.

EMODnet collects, validates and provides access to relevant marine chemistry data to assess the state of ecosystems in accordance with the Marine Strategy Framework Directive. The Chemistry component of EMODnet has adopted and adapted SeaDataNet standards and services, and delivers regional aggregated datasets receiving additional quality control of metadata and data ([www.emodnet-chemistry.eu/products](www.emodnet-chemistry.eu/products)). These *in situ* observation collections contain oxygen, nitrate, phosphate, silicate and ammonium profiles all in µmol $l^{-1}$ and Chl-a profiles in mg Chl $m^{-3}$. The North-East Atlantic Ocean dataset includes data from the OVIDE section between Portugal and Greenland in spring 2010 and data from the springtime PELGAS cruises on the Bay of Biscay. For the Mediterranean Sea dataset, only Chl-a is presented as it has the best spatial cover as compared to other variables.

BGC-Argo floats are autonomous profiling floats advected by currents (Biogeochemical-Argo Planning Group, 2016). These floats acquire vertical profiles of temperature, salinity and key biogeochemical variables over complete seasonal cycles. In this study, we use the vertical profiles of dissolved oxygen, nitrate (both estimated in µmol $kg^{-1}$ and converted in µmol $l^{-1}$) and Chl-a concentrations (in mg Chl $m^{-3}$) collected with 2 BGC-Argo floats in the IBI region. The first float is an APEX profiler (World Meteorological Organization (WMO) number 5904479), deployed in the North Atlantic Ocean by the University of Washington (Seattle) in February 2014 and which provided biogeochemical measurements until December 2016. The second float is a PROVOR-II profiler (WMO number 6901648), deployed in the Western Mediterranean Sea by the French Villefranche Oceanographic Laboratory in July 2014 and recovered in May 2016. The float data can be downloaded from the Argo Global Data Assembly Centre in France ([ftp://ftp.ifremer.fr/ifremer/argo](ftp://ftp.ifremer.fr/ifremer/argo); Carval, et al., 2017). The CTD and trajectory data are quality-controlled following Wong et al. (2015). The raw BGC signals are transformed into Chl-a, oxygen and nitrate concentrations following Schmechtig et al. (2015), Thierry et al. (2016) and Johnson et al. (2016), respectively. Finally, corrections are applied on each variable to reduce calibration biases and sensor drifts. For the APEX float observations, the three variables are "delayed mode" data, and are adjusted following Johnson et al. (2017). For the PROVOR float observations, oxygen and nitrate are "Real time" data and Chl-a is "adjusted" data, they are adjusted following Mignot et al. (2018), and the first five months of nitrate measurement were masked due to spurious values.

## 4. IBI36 evaluation

The skills of the pre-operational qualification simulation are evaluated by comparing model results for the main biogeochemical variables (Chl-a, NPP, nutrients and oxygen) to satellite derived estimations and *in situ* observations between 2010 and 2016. In function of data availability, daily to seasonal time scale is evaluated. The spatial distribution (two-dimensional longitude-latitude plots), time series, vertical profiles and statistics performance are presented, using GODAE-like metrics (Hernandez et al., 2009, 2015), in order to assess the quality of the PISCES biogeochemical component in terms of consistency and quality/accuracy. The GODAE "Class 1" metrics are a direct comparison to observed quantities and give a general overview of the model's ability to be consistent with the general features of the IBI European waters. The GODAE "Class 4" metrics provide a series of statistics and quantify the differences between model and observations at their location and time.

### 4.1 Satellite derived estimations

For comparison to the satellite derived estimations (Chl-a and NPP), the model is interpolated onto the data grid. Satellite estimates are scarce north of 50°N during the winter season, especially between November and February due to omnipresent

cloud coverage that dramatically limits the observation of Chl-a concentrations. Consequently, the model outputs (Chl-a and NPP) are masked based on data availability, thus the annual average is done on the same number of samples. The annual average is calculated using the 7 years of simulation, from 2010 to 2016. Time series, Hovmöller diagrams and time correlation are based on monthly averages. The time series are presented for several small boxes defined and presented in Section 2 and Fig. 1a. Some of them are located offshore to the open ocean (boxes 1 to 4, 11 and 12) and the others follow the coastal areas (boxes 5 to 10).

### 4.1.1 Chlorophyll-a

The model sea surface Chl-a concentration is compared to the ESA OC-CCI product. The annual average Chl-a spatial distribution, the bias and the root mean square error (RMSE) are presented in Fig. 2. The time evolution of Chl-a at 15°W longitude (Hovmöller diagram) is on Fig. 3 in order to discuss the seasonal dynamics of the North Atlantic part. Time series for the 12 boxes already introduced as well as the spatial distribution of the temporal correlation at each grid point are presented in Fig. 4. Global statistics are synthesised on Table 1.

The averaged Chl-a over the IBI domain ($0.615 \pm 0.69$ mg Chl m$^{-3}$) is close to the ESA OC-CCI product ($0.555 \pm 0.63$ mg Chl m$^{-3}$) resulting in a low percent bias of 10.8% and a high correlation of 0.81 (Table 1). The large-scale distribution of Chl-a is correctly reproduced: the North Atlantic subtropical gyre with low surface concentrations ($< 0.1$ mg Chl m$^{-3}$), increasing concentrations when moving to the north and the highest values on the continental shelf. The Chl-a signature of the shelf-slope front is well marked west of the British Isles to the Faroe-Shetland Channel. The maximum coastal Chl-a is supplied by nutrient input from rivers, resuspension by strong tidal currents in the Northern shelf and upwelling off the Iberian and Moroccan coasts.

Major biases are located on the continental shelf (Fig. 2c). The model simulates a higher annual average in the northern part (southern North Sea, English Channel, Irish Sea and Faroe Islands), the French coasts of the Bay of Biscay and the ROFI of the Ebro and Rhone rivers. The model underestimates Chl-a concentrations off the coast of Morocco (south of Agadir) and in the region linking the North Sea to the Baltic Sea (Kattegat and Skagerrak) (Fig. 2c). The spatial distribution of the RMSE (Fig. 2d) between the simulation and the satellite product is comparable to the annual average of Chl-a (Fig.2 a and b). RMSE increases from south to north in the North Atlantic part and is the highest in coastal areas (Fig. 2d).

The seasonal dynamics of the North Atlantic spring phytoplankton bloom, expressed as Chl-a, is depicted by the Hovmöller diagram at 15°W (Fig. 3) and the time series in boxes 1 to 3 (Fig. 4). In the subtropical North Atlantic, Chl-a concentrations are limited throughout the year. A moderate Chl-a peak develops in March in the southern part of the domain (peak to 0.4 mg Chl m$^{-3}$ in box 3) and gradually moves northward while intensifying (peak to 1.3 mg Chl m$^{-3}$ in box 1). The bloom onset is well reproduced in the south (r = 0.91 in box 3), but it spreads more rapidly to the north. The observed peak reaches Iceland in summer (June-July) while the simulated peak reaches Iceland in May. The summer decrease after the bloom is then earlier and sometimes more pronounced in the model, explaining the alternation of positive and negative biases in the Hovmöller diagram at 15°W (Fig. 3c), the increasing RMSE from south to north (Fig. 2d) and the lower temporal correlation to the north of the domain (Fig. 4). The south part has limited seasonal variations while the north part shows a strong seasonal cycle. The ESA OC-CCI product also highlights a large inter-annual variability in the north part while the model seems to be dominated by the seasonal dynamics (Fig. 3 and 4). But one part of the signal is however missing due to cloud cover masking several months each winter.

In the southern half of the IBI domain, south of 50°N in the Atlantic part and in the Mediterranean, the simulated seasonal cycle of Chl-a is in phase with satellite product in view of the low RMSE (Fig. 2d) and high temporal correlation (Fig. 4). Coastal ecosystems of the Bay of Biscay (box 7) show a peak biomass during spring bloom, while the upwelling off Portugal and Morocco (boxes 8 and 10) presents a maximum in spring with more interannual variability off Morocco. In the Gulf of

Cadiz (box 9) and the Western Mediterranean (boxes 11 and 12), IBI36 succeeds in reproducing the seasonal cycle of Chl-a (Fig. 4), with a high correlation coefficient ($r > 0.71$) with the satellite product.

In shallow Northern seas, the model does not match satellite product (boxes 4 to 6 in Fig. 4). In the open North Sea (box 4),
the first peak is usually reproduced, but the data present a strong interannual variability. In the southern North Sea (box 5) and the English Channel (box 6), model and data are dominated by the seasonal dynamics. The spring bloom is in phase but high Chl-a concentrations persist in summer in the model, while remote sensing estimates show a sharp decrease after the spring bloom. These coastal regions present the highest biases, highest RMSE and low temporal correlation.

### 4.1.2 Net primary production

Simulated depth integrated NPP is compared to the three NPP products (VGPM, Eppley-VGPM and CbPM). Figure 5 presents the annual average distribution for the simulation and the mean of the three NPP products, the standard deviation of the three NPP products and the bias between the simulation and the mean of the three NPP products. For time series (Fig. 6), the three products are presented separately because they do not all have the same seasonal behaviour, and therefore an average would prevent any analysis. Global statistics are synthesised on Table 1.

On annual average, the IBI36 system provides for a NPP of 230 gC m$^{-2}$ yr$^{-1}$ at the western boundary of the domain that gradually increases towards the coasts (Fig. 5b). The highest NPP (1700 gC m$^{-2}$ yr$^{-1}$) is found in the coastal regions of the North Sea, where rivers and mixed water columns supply the euphotic layer with nutrients. Compared to the mean of the three NPP products (Fig. 5a), the large-scale distribution is reproduced. The cross-shore gradients are reproduced; the signature of the shelf-slope front west of the British Isles to the Faroe-Shetland Channel is captured. However the IBI36
system underestimates the NPP by a factor of 1.5 on average over the domain. The most important differences concern the Kattegat/Skagerrak area and the Norway current, with a factor of 3-4. On the other hand, it should be noted that the dispersion between VGPM, Eppley-VGPM and CbPM products is considerable (Fig. 5c). Except to the open boundaries, the bias of the model equals almost the variability of the NPP products.

The time series confirm the considerable spreading between the three NPP products (Fig. 6). The simulated NPP is generally
in line with the two VGPM-based products (VGPM and Eppley-VGPM) with a time correlation higher than 0.7 in a majority of boxes. The very good correlation in the south part of the Atlantic (box 3, $r = 0.91$ with the VGPM) decreases northward (boxes 2 and 1), as IBI36 produces a moderate and above all an earlier production peak. The behaviour of the NPP in the North Atlantic part is consistent with the seasonal dynamics of the sea surface Chl-a described in the previous section. The coastal waters of the Northern Seas and Atlantic part as well as the Mediterranean (boxes 4 to 12; Fig. 6) also show a
simulated seasonal cycle of NPP close to the VGPM-based products. On the other hand, the correlation is low compared to the CbPM product, but this latter delivers a seasonal cycle generally very different from the VGPM-based products. The CbPM signal is sometimes in phase opposition with IBI36 (boxes 3, 5, 9, 10 and 11) while the comparison with VGPM results in high correlation in these same boxes.

The averages simulated NPP ($441.7 \pm 203.5$ gC m$^{-2}$ yr$^{-1}$) is close to CbPM ($518.1 \pm 660.9$ gC m$^{-2}$ yr$^{-1}$) and twice lower than
VGPM ($871.5 \pm 557.2$ gC m$^{-2}$ yr$^{-1}$) (Table 1). But spatial distribution is better correlated with the VGPM-based products (Table 1). The VGPM is the most productive product, with a marked cross-shore gradient and the highest seasonal amplitude (Fig. 6). The Eppley-VGPM behaves the same way as the VGPM, but less productive (Fig. 6). The CbPM is the less productive, with a poorly marked cross-shore gradient, the lowest coastal production and a less pronounced seasonal cycle, sometimes out of phase with VGPM-based products (Fig. 6). A few extreme values near rivers increase the averaged NPP of
CbPM and give rise to a high standard deviation.

In summary, IBI36 provides an averaged NPP similar to CbPM. The spatial distribution, cross-shore gradients and seasonal variations are generally in good agreement with the VGPM-based products, but IBI36 is half as productive (mean factor of 1.5). The modelled NPP is thus within the range of variability of the satellite derived estimates.

**4.2 *In situ* historical data**

In the following, the simulation is compared to ICES and EMODnet *in situ* historical databases using daily averaged model outputs. ICES data are mainly located in the shallow and coastal waters of the Northern seas (Fig. 1b). EMODnet regional datasets cover the North-East Atlantic Ocean and the Western Mediterranean Sea. Global statistics are summarised in a Taylor diagram (Fig. 7).

**4.2.1 Northern Seas**

Shallow Northern seas are assessed using oxygen, nutrients and Chl-a from ICES database. Dispersion diagrams for the full set of match-ups are presented on Fig. 8. Sea surface spatial distribution and seasonal cycle are on Fig. 9 and 10.

The oxygen match-ups are well aligned along the bisector with a good correlation (r = 0.77) and a normalized standard deviation of 0.91 indicating that the model reproduces the amplitude and variability of the observations (Fig. 7 and 8). Temporal evolution of sea surface concentrations shows the realistic amplitude and phase (Fig. 10). Sea surface oxygen is

slightly overestimated in the North Sea and English Channel, with an average bias of 10.7 µmol l$^{-1}$, corresponding to a percent bias of 4% (Fig. 9). In addition, the model does not capture the lower sea surface oxygen concentrations measured during 2014-2015 period (Fig. 10). This anomaly is located in the region linking the North Sea to the Baltic Sea (Kattegat and Skagerrak), the eastern open boundary of the domain. But no reference to this event has been found in the literature.

The distribution of nitrate also follows the bisector, with a noticeable dispersion (Fig. 8) which deteriorates the statistics

(Fig. 7). The model generally underestimates sea surface nitrate with an average bias of -1 µmol l$^{-1}$ (9.6% percent bias) (Fig. 9). The time series shows a seasonal cycle in phase, but excessive nitrate concentrations are simulated in spring and summer when the observed concentrations are very low (Fig. 10). Very high values of 100 to 300 µmol l$^{-1}$ are simulated throughout the year in the vicinity of river flows between the Rhine and Elbe and impact the time series.

Phosphate and silicate are overestimated for low concentrations during spring-summer seasons, while higher concentrations

during winter conditions are better captured (Fig. 8 and 10). The phosphate dispersion diagram shows two high-density zones. The spring-summer overestimation is mainly along the coasts. Winter conditions are better captured, although still a little high. The data show a marked seasonal cycle while simulated phosphate levels remain too high throughout the year. The average bias of 0.22 µmol l$^{-1}$ or 48.3 % percent bias is reduced to 31.5% when the pathway to Baltic Sea is excluded. Silicate has an average bias of 2.1 µmol l$^{-1}$ or 46.8% percent bias. They are slightly overestimated in the open North Sea and

underestimated along the coasts between the Rhine and the Elbe. In addition, percent bias decreases to 30.8% when the pathway to Baltic Sea is excluded. Ammonium shows a high dispersion but the magnitude is captured. The model does not reproduce the variability observed in data (Fig. 8e), and the seasonal cycle is out of phase (Fig. 10e). The statistics give thus poor performances for phosphate and silicate, even outside the Taylor diagram for ammonium (Fig. 7).

Chl-a provides a satisfying spatial distribution (Fig. 8) but mean Chl-a concentrations along the coasts are underestimated

(Fig. 9). The seasonal cycle is captured, although the model predicts a slow spring increase instead of a strong bloom in mid-march (Fig. 10). Coastal Chl-a appears to be underestimated compared to ICES *in situ* data, while overestimated compared to satellite estimates (see Sect. 4.1.1 and Fig. 2). The statistics are not satisfying (Fig. 7) while the density plot, surface distribution and time series (Fig. 8 to 10) give a quite positive evaluation.

Statistics are not really rewarding (Fig. 7) because they are strongly degraded by extreme values at the mouth of rivers or

highly targeted areas such as Kattegat/Skagerrak. They alone do not allow understanding the characteristics of the IBI36 system. To interpret and complete the statistics, the mean spatial distribution and daily averaged temporal evolution are necessary. These usually give a more positive assessment because extreme values are filtered (see the details in the legend of Fig. 9 and 10). Oxygen is the best performing variable in the Northern Seas, and its satisfying statistics allow deepening the analysis of oxygen match-ups between ICES and IBI36.

Oxygen content is a key element in biogeochemical cycles and can be an indicator of the health of marine ecosystems: for this reason the minimum oxygen concentrations are now analysed. For that, the absolute minimum is extracted for each pixel of ICES and collocated IBI36. The lowest concentrations are located in the eastern part of the North Sea (Fig. 11a). The minimum remains high in winter while it sharply decreases or even reaches anoxic conditions in summer (Fig. 11b). The minimum reported by ICES remain lower than usually during 2011 and 2015 winters, but they come from a few

measurement points very close to the coast in the vicinity of river mouth, not captured by the IBI36 system. The spatial distribution of the simulated minimum as well as its seasonal evolution is consistent with the data (r = 0.77). But please remind that ICES data only permit identifying the North Sea because the data density strongly decreases outside. So extending this analysis to the full set of simulated oxygen over the IBI domain (not only the match-ups with ICES), IBI36 also simulates minimum levels in the Celtic Seas, Armorican shelf, coastal areas of Scotland and Western Ireland. Ciavatta

et al. (2016) and the OSPAR commission point out these aforementioned regions as eutrophication problem areas and Breitburg et al. (2018) also report low and declining oxygen levels in almost all coastal waters of the North-West European shelf.

Continental shelf areas vulnerable to oxygen deficiency were estimated by Ciavatta et al. (2016), considering vulnerable area when at least one daily value is below the 6 mg l$^{-1}$ (187.5 µmol l$^{-1}$) threshold during the time of the simulation. Using the

same method as Ciavatta et al. (2016), the IBI36 system predicts a maximum surface area exposed to oxygen deficiency of 280 000 km$^2$. The vulnerable surface area is almost non-existent in winter because waters are well oxygenated due to strong mixing and extends to an average surface area of 85 000 km$^2$ in summer (Fig. 11c), associated with deoxygenated waters that can reach anoxic conditions in the North Sea and along the west coasts of France.

### 4.2.2 North-East Atlantic waters

The North-East Atlantic part is evaluated using the EMODnet regional dataset. Global statistics are very satisfying (Fig. 7) as *in situ* measurements cover the entire water column. However, performance between the vertical and sea surface distribution differs greatly. To illustrate this contrasting performance, comparison to OVIDE section and PELGAS data are detailed below using Fig. 12 and 13.

The OVIDE radial section sampled in June 2010 between Portugal and Greenland (Fig. 12) illustrates the vertical

distribution of biogeochemical tracers in the open Atlantic. Model oxygen and nutrients show very good statistics with OVIDE data, with coefficient correlation higher than 0.95 (Fig. 7). The dispersion diagram for oxygen shows two pools of high density: one for low oxygen values, and the other one for high concentrations. Throughout the OVIDE vertical section, the minimum oxygen level is around 1000 meter deep. Low oxygen content in the eastern part of the section is due to MOW on the shelf of the Iberian Basin. Oxygen maximum around 2500 meters relies on recently ventilated Labrador Sea Water

(Garcia-Ibanez et al., 2015) that reaches the western part of the section. The three nutrients present a maximum around 1000 meters, the lower values at this depth being due to MOW. High silicate (45-50 µmol l$^{-1}$) near the bottom reflects the influence of Antarctic Bottom Water in the North-East Atlantic Ocean (Garcia-Ibanez et al., 2015). However, vertical profiles of oxygen are somewhat smoothed. The minimum and maximum at respectively 1000 and 2500 meters are not pronounced enough, resulting in a normalised standard deviation of 0.74 (Fig. 7). Nutrient profiles are also smoothed, but

this is less visible (normalised standard deviation close to 1; Fig. 7) than on oxygen, as the latter has much stronger vertical gradients.

The PELGAS spring data (Doray et al., 2018a, 2018b) are used to illustrate the mean sea surface distribution in the Bay of Biscay for spring conditions (Fig. 13). Surface statistics are significantly degraded compared to vertical statistics (Fig. 7). Simulated sea surface oxygen concentrations present an average bias of +16.4 µmol l$^{-1}$, which corresponds to a percent bias

of 6.3% (Fig. 13). Nutrient distribution is realistically simulated, except at the ROFI of French rivers. Surface oxygen bias and excessive nutrient discharges were already highlighted in the Northern Seas using the ICES database. The mean surface

Chl-a distribution is similar to the data: the cross-shore gradient is realistic with concentrations of 0.3 mg Chl m$^{-3}$ offshore, which increase to 6 mg Chl m$^{-3}$ along the French coast.

### 4.2.3 Mediterranean Sea

The Mediterranean Sea is assessed using EMODnet regional dataset that has a very good spatial coverage for oxygen and Chl-a tracers, while nutrient data are limited to the northern part of the domain. Oxygen comparison gives the same conclusions as for the North Sea and the Atlantic: the model succeeds in reproducing the amplitude and variability of oxygen, but a constant bias persists. So only the sea surface Chl-a distribution is presented here (Fig. 14). High coastal values are located along the Catalan coast, in the ROFI of the Ebro, along the Costa Blanca and along Algeria. Two highly
productive areas are located further offshore: one in the convection zone of the Gulf of Lions and the other in the Algerian Basin between Sardinia and Algeria. Everywhere else, Chl-a is lower. The model simulates Chl-a higher than EMODnet in the Alboran Sea and in the ROFI of the Rhône River. But in a general way, the model reproduces the mean spatial distribution of surface Chl-a in the Mediterranean.

### 4.3 BGC-Argo data

The free-drifting BGC-Argo profiling floats allow continuous monitoring of dissolved oxygen, nitrate and Chl-a of the upper 1000 meters of the ocean. 2 BGC-Argo floats are used, one in the North Atlantic Ocean and the other in the Western Mediterranean Sea in order to discuss the model quality in reconstructing the seasonal vertical dynamics, and the key coupled physical-biogeochemical processes. Density plots between the BGC-Argo data and simulated fields are presented on Fig. 15 and time evolution of the vertical profiles of oxygen, nitrate and Chl-a along the float trajectory are on Fig. 16. The
quantitative comparison is summarized by the statistics of Fig. 7.

Overall, the model predictions are in good agreement with the BGC-Argo observations with correlation coefficients greater than 0.8 for oxygen and nitrate profiles (Fig. 7). The model tends to overestimate low concentrations and underestimate high concentrations of oxygen and nitrate as shown by the distribution of match-ups which deviate from the bisector (Fig. 15). Time evolution of the vertical profiles in Fig. 16 shows that the deep oxygen minimum and nitrate maximum are not
pronounced enough in the model. Oxygen remains 20 µmol l$^{-1}$ too high and nitrate 2 µmol l$^{-1}$ too low. The smoothing of the vertical profiles of oxygen and nutrients was already highlighted by the comparison to OVIDE.

The IBI36 system succeeds in reproducing the winter vertically mixed water column that enriches the first few hundred metres of the water column with oxygen and supplies the surface with nutrients. Seasonal re-stratification and the shoaling of the MLD trigger the onset of the spring phytoplankton bloom (Fig. 16). In the Atlantic Ocean (Fig. 16, left side), the MLD
reaches 400-500 m depth during winter. Depth of the ventilation has a clear interannual variability, as shown by the deeper mixing during winter 2015 with respect to the following year. This ventilation also enriches the surface in nutrients. If winter processes are well reproduced, the onset of the simulated bloom is however too early. The intensity of the bloom is misrepresented in the model as surface Chl-a concentrations remain significantly lower than BGC-Argo data during the spring bloom, decrease rapidly after the bloom and remain at a low level during summer. This behaviour as regards of the
BGC-Argo data is consistent with the comparison to ESA OC-CCI ocean colour product in box 2 of Fig. 4. The time evolution of vertical profiles highlights that the high surface Chl-a associated with the spring bloom migrate to the sub-surface during the stratified season in the model while they remain at the sea surface in the data. Indeed, a Deep Chl-a Maximum (DCM) develops in summer in the model simulation while in the observations, the Chl-a maximum is able to maintain at the surface during summer.

In the Mediterranean Sea (Fig. 16, right side), the seasonal cycles of Chl-a and oxygen are characterized by the formation of a DCM (Mignot et al., 2014; Lavigne et al., 2015), which typically establishes during the stratified season. The DCM is associated to a Deep Oxygen Maximum (DOM) at the layer of the DCM due to intense phytoplankton production during

spring and summer (Estrada et al., 1985). These maxima are also associated with the limit between nutrient-depleted and the nutrient-rich layers, termed nutricline (Estrada et al., 1993). The model correctly reproduces the time evolution of the nutricline, as well as the temporal evolution, vertical displacement and intensity of the DCM and DOM. The IBI36 system compares well with the Mediterranean float to reproduce the vertical dynamics of the phytoplankton chlorophyll and oxygen, suggesting that the seasonal succession of physical-biogeochemical processes is captured.

## 5 Discussion

An extended validation of the pre-operational qualification simulation has allowed understanding the strengths and weaknesses of the biogeochemical component of the IBI36 system, providing the trails for improvement to be explored, which are here discussed.

Mismatches between simulated and satellite derived estimations of Chl-a and NPP increase when approaching the continental shelf. The uncertainties of the modelled Chl-a with respect to ESA OC-CCI product are determined by calculating the bias and RMSE. Highest uncertainties are located in coastal areas, and can be explained by temporal discrepancies between the simulation and ESA OC-CCI product. For NPP, the uncertainties are apprehended by the comparison of the standard deviation of the three NPP products and the bias between the simulation and the mean of the three NPP products. Bias of the model is included in the standard deviation of the NPP products. The modelled NPP is then included within the range of uncertainty of the satellite derived products.

Continental margins are very productive regions and play an important role in the biogeochemical cycle of nutrients and carbon. They are the site of complex interactions between physical, chemical and biological processes that include exchanges between shelf and the open ocean, sediment-water interactions, air-sea fluxes, and land-ocean freshwater inputs. In addition, coastal systems are locally strongly affected by human activities. All these interactions make the continental shelf a challenge to obtain realistic models.

Continental margins are also the areas where the uncertainties of satellite products are the greatest. Coastal areas are complicated areas for satellite sensors to measure due to interference from Chl-a content with other optically absorbing elements such as suspended matter, coloured dissolved organic matter and bottom reflectance, resulting in a 100% uncertainty in the estimate of Chl-a, compared to 30% for the open ocean (Moore et al., 2009). A good example is given in the North Sea example, where the model underestimates coastal Chl-a with respect to the ICES *in situ* data (Sect. 4.2.1), which however appears in contrast with the overestimation with respect to the satellite ESA OC-CCI product (Sect. 4.1.1). The dispersion between the three NPP products is also considerable. Campbell et al. (2002) pointed out that the "best-performing algorithms generally fall within a factor of 2 of the estimates derived from [14]C", and that NPP products have poor performances for water columns with depths less than 250 meters (Saba et al., 2011). Schourup-Kristensen et al. (2012) also reported that the VGPM product is twice as productive as biogeochemical models along the European coasts. This high uncertainty in NPP products prevents a quantitative assessment. Additionally, they do not have the same seasonal dynamics; CbPM has a seasonal cycle distinct or even out of phase from the two others NPP models in a major part of the domain (Sect. 4.1.2). An extensive dataset of measures of primary production in the IBI European waters would be necessary to evaluate the three NPP products and deepen the analysis.

As mentioned above, continental margins are very sensitive to the boundary conditions of the model such as air-sea interactions, river inputs, water-sediment interactions but also open boundary conditions. IBI36 performances decrease as these limits approach, as detailed below:

1/ Oxygen concentrations at the sea surface are very sensitive to ocean-atmosphere exchanges, as ocean oxygen balances with oxygen from the atmosphere within a few weeks. The slight overestimation (around 4-6%) of the IBI system is not yet fully understood. Solubility of modelled oxygen is similar to the one from ICES and EMODnet *in situ* historical databases

(not shown), suggesting that biases in sea surface temperature or salinity can not explain the biases in sea surface oxygen.
The other key components (gas transfer velocity and biological production) need to be further explored to better understand this overestimation and reduce it.

2/ The continental shelf ecosystem is strongly driven by river discharges, especially in the Northern seas. The seasonal cycle of phosphate and silicate is not sufficiently marked, and the spring bloom is not intense enough as the one reported in ICES data. In the coupled IBI36 system, nutrient inputs at river points are prescribed using annually averaged values, while inputs usually follow a seasonal cycle related to precipitation and watershed erosion. The increased discrepancies when approaching the coasts are related to a poor representation of river nutrient discharges due to a crucial lack of available measurements. A time evolution or at least a seasonal variation would be necessary to apprehend the phytoplankton dynamics in the coastal areas triggered by rivers plume events.

3/ Permanent burial in sediments is not considered in the IBI36 system because strong tidal currents prevent organic matter from reaching the bottom and accumulating in the sediments. This assumption may be too restrictive for the whole model domain. The future system will adjust the efficiency of permanent burial based on bottom friction. But the fact remains that the treatment applied to the ocean floor remains very basic in the standard version of PISCES. A sediment module that takes into account biogeochemical processes in sediments and at the sediment-water interface is most certainly required for the IBI configuration where the continental shelf covers a large area.

4/ Open boundary conditions are also fundamental. A perfect example is in the Kattegat/Skagerrak area, connecting the North Sea to the Baltic Sea, at the eastern boundary of the IBI domain. The statistics for the Northern Seas are strongly affected by extreme values at the mouth of rivers (discussed above) and by the highly targeted area that Kattegat/Skagerrak area is. The CMEMS Baltic Sea regional configuration instead of the global product should be tested at this eastern open boundary of the IBI36 system.

Moving away from the continental margins, statistics are very satisfying for oxygen and nutrients, as the entire water column is considered. The model performs in reproducing the vertical structure of oxygen and nutrients but the profiles appear too smoothed. The deep minima and maxima are not pronounced enough. This behaviour is also observed in the global model at ¼° (Perruche et al., 2016) used to set-up the initial and open boundary conditions, and can originate from the physical or biogeochemical models. Different approaches are currently under study: in particular, vertical diffusion could explain the loss of peaks and minima in vertical profiles, but biogeochemical processes (e.g., parameterization of remineralisation processes, rate of sinking of particulate detritus, vertical migration of zooplankton which export organic matter at depth) will also be investigated.

Finally, BGC-Argo floats allowed better understanding the phytoplankton and oxygen vertical dynamics. The Mediterranean Sea dynamics is well captured by the model, in terms of timing, vertical migration of the maximum chlorophyll and the formation of an oxygen maximum linked to the DCM. In the Atlantic part, winter processes are captured but the bloom onset is early. In fact, the onset of the spring bloom is correct in the south part of the domain, but it spreads more rapidly to the north (see the comparison to satellite data; Sect. 4.1.1). The summer decrease after the bloom is then earlier and sometimes more pronounced in the model, and the Chl-a maximum migrates to the sub-surface during summer with the formation of a DCM while maximum Chl-a remains at the surface in BGC-Argo estimates. Indeed, once the spring bloom is over, PISCES cannot maintain the phytoplankton on the surface, there is always an element that becomes limiting at the end of spring (oligotrophic conditions). This behavior is also present in the global model (Perruche et al., 2016, 2018). In fact, it is not clear how biological production can be maintained at the surface throughout the summer with low nitrate content as observed in BGC-Argo (Fig. 16). However, the analysis is limited to only one float measuring simultaneously oxygen, nitrate and Chl-a in the North-East Atlantic part of the IBI domain. Additional floats are essential to understand the seasonal dynamics of phytoplankton, oxygen and nitrate and better apprehend the involved physical-biogeochemical coupled processes.

**6 Conclusions**

In the framework of CMEMS, the IBI-MFC Team has developed an operational system in order to monitor and forecast the ocean dynamics and marine ecosystems of the IBI European waters. A 7-year pre-operational qualification simulation (2010-2016) delivers the initial conditions to the analysis and forecast system. This paper provides an extended validation of this pre-operational qualification simulation in order to evaluate the capacity of the IBI36 system to reproduce the surface and vertical distributions, as well as seasonal cycles of the main biogeochemical variables (Chl-a, NPP, nutrients and oxygen) using GODAE-like metrics. The different kinds of metrics (direct comparison and statistics) are necessary and complementary in order to have a complete description of the model's performance in terms of consistency and quality/accuracy. This paper represents the first validation of the biogeochemical component of the IBI36 system: the objective is to show that PISCES can be used for operational applications, and that it is a suitable tool at such a resolution. Chl-a and NPP are compared to satellite estimates, describing here their mean spatial distribution and seasonal cycle. Oxygen, nutrients and Chl-a concentrations are compared to *in situ* observations from ICES, EMODnet and the BGC-Argo float network, using daily averages of the model outputs. Observational data are available for the Northern Seas, the North-East Atlantic waters and the Western Mediterranean, and allow evaluating the vertical distribution as well as shallow and coastal distributions. Some of these areas are outside of the IBI Service Domain (that is the geographical domain covered by the CMEMS IBI-MFC products), but in order to take advantage of their *in situ* observational coverage, we evaluated the IBI Extended Domain. Main results are here summarized:

- The mean distribution and the seasonal cycle of sea surface Chl-a is in line with satellite estimates, particularly south of 50°N in the Atlantic and the Mediterranean. The BGC-Argo floats suggest that the seasonal succession of vertical physical-biogeochemical processes is well captured by the model in the Mediterranean, with the development of a seasonal DCM below the MLD. On the other hand, on the North-East Atlantic waters, the spring phytoplankton bloom spreads more rapidly to the north and Chl-a maximum is not able to maintain at the surface during the stratified season, and migrates to the sub-surface instead. The BGC-Argo floats, although their spatial coverage is limited, opens new doors to understand and improve the seasonal dynamics of phytoplankton in the models.

- NPP is a complex field to evaluate, as satellite derived products give widely different estimates among themselves. The model averaged spatial distribution is close to CbPM product, but spatial distribution, cross-shore gradients and the seasonal variations are better correlated to the VGPM-based products. The modelled NPP is thus within the range of variability of the satellite derived estimates.

- Vertical distribution of oxygen and nitrate obtains very good statistics. The amplitude and variability of the observations are captured by the model, but the vertical profiles of oxygen and nitrate appear somewhat smoothed in the wider ocean.

- The continental shelf area shows the highest biases in nutrients and Chl-a as river nutrient discharges and sedimentary processes strongly influence the seasonal cycle of nutrients and thus phytoplankton dynamics.

- The continental shelf area of IBI domain appears vulnerable to oxygen deficiency; especially the wide continental shelf covering the Northern seas and the Bay of Biscay. Maximum surface area can reach 280 000 km$^2$ during the time of the simulation, but the mean seasonal extension varies from a very restricted surface area in winter to 85 000 km$^2$ in summer.

This extended evaluation has allowed understanding the strengths and weaknesses of the biogeochemical component in the IBI36 system. The pre-operational qualification simulation performs in reproducing the main biogeochemical characteristics of IBI European waters. PISCES is then a suitable tool at such a resolution and can be used for operational analysis and forecast applications. Future improvements were also explored. Finally, the operational analysis and forecast IBI36 system can be a useful tool to better understand and monitor the health of marine ecosystems (von Schuckmann et al., 2016, 2018).

**Code availability**

The IBI36 configuration is based on the NEMO 3.6 version developed by the NEMO consortium. NEMO modelling system is freely available at http://www.nemo-ocean.eu. The biogeochemical model PISCES v2 is part of the NEMO modelling platform and is available via the NEMO web site. Model initialization and boundary conditions are available via CMEMS (http://www.marine.copernicus.eu).

**Data availability**

The regional ESA Ocean Colour CCI product for the North Atlantic and Arctic Oceans with a resolution of 1 km is distributed via CMEMS (http://www.marine.copernicus.eu). Primary production products are distributed by the Oregon State University (www.science.oregonstate.edu/ocean.productivity). The International Council for the Exploration of the Sea (ICES) oceanographic database is available at www.ices.dk/marine-data/data-portals. The European Marine Observation and Data Network (EMODnet) is available at www.emodnet-chemistry.eu/products. EMODnet regional aggregated datasets are generated by EMODnet Chemistry under the support of DG MARE Call for Tenders MARE/2008/03-lot3, MARE/2012/10-lot4 and EASME/EMFF/2016/006-lot4. The North-East Atlantic Ocean regional dataset is aggregated, standardized and quality controlled by IFREMER / IDM / SISMER - Scientific Information Systems for the SEA (2018) from France, and the Mediterranean Sea dataset by Hellenic Centre for Marine Research, Hellenic National Oceanographic Data Centre (HCMR/HNODC) (2018) from Greece. Biogeochemical-Argo (BGC-Argo) float data can be downloaded from the Argo Global Data Assembly Centre in France (ftp://ftp.ifremer.fr/ifremer/argo).

**Author contribution**

Elodie Gutknecht (Mercator Ocean) contributed to the set-up of the IBI36 system, performed the evolution of model outputs by comparison to satellite and in-situ datasets, prepared the figures and is the main writer of the manuscript. Guillaume Reffray (Mercator Ocean) has designed and developed the IBI36 physical-biogeochemical coupled system and has performed the simulations. Alexandre Mignot (Mercator Ocean) provided corrected BGC-Argo dataset. Tomasz Dabrowski (Marine Institute) is responsible of the evaluation and validation of the reanalysis and analysis/forecast systems in the CMEMS IBI-MFC. Marcos Garcia-Sotillo (Puertos del Estado) has the leadership responsibility of the CMEMS IBI-MFC.

**Acknowledgments**

The authors acknowledge financial support through CMEMS. CMEMS is implemented by Mercator Ocean International in the framework of a delegation agreement with the European Union. They thank their colleagues of Mercator Ocean for their contribution to the model development and evaluation (Bruno Levier, Mounir Benkiran). We would also like to thank the PISCES community, and especially Olivier Aumont, for all the constructive discussions. Finally, the authors are grateful to the two anonymous reviewers for their relevant and constructive comments.

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

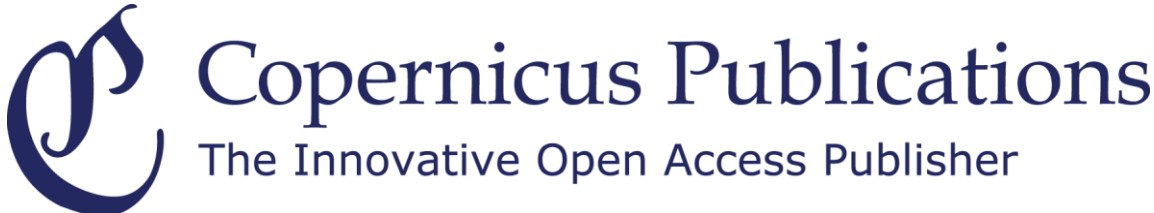

**Figure 1: The logo of Copernicus Publications.**



Table 1: Synthesis table for Chl-a (mg Chl m$^{-3}$) and NPP (mg C m$^{-2}$ d$^{-1}$) assessment against satellite derived estimations. Mean and standard deviation, Mean Error $= \langle (model - obs) \rangle$, RMSE $= \sqrt{\langle (model - obs)^2 \rangle}$, Percent Bias (%) $=$ |$Mean\ Error/Mean\ Obs$| and correlation are computed for the IBI Extended Domain, using model and observations averaged over the length of the simulation (2010-2016).

| Variable | Dataset | Mean ± std | Mean Error | RMSE | Percent Bias (%) | Correlation |
|---|---|---|---|---|---|---|
| **Chl-a** | IBI36 | 0.615 ± 0.69 | | | | |
| | ESA OC-CCI | 0.555 ± 0.63 | 0.06 | 0.42 | 10.8 | 0.81 |
| **NPP** | IBI36 | 441.7 ± 203.5 | | | | |
| | VGPM | 871.5 ± 577.2 | -429.8 | 636.7 | 49.3 | 0.65 |
| | Eppley | 557.5 ± 358.3 | -115.8 | 295.3 | 20.8 | 0.66 |
| | CbPM | 518.1 ± 660.96 | -76.4 | 602.78 | 14.7 | 0.45 |

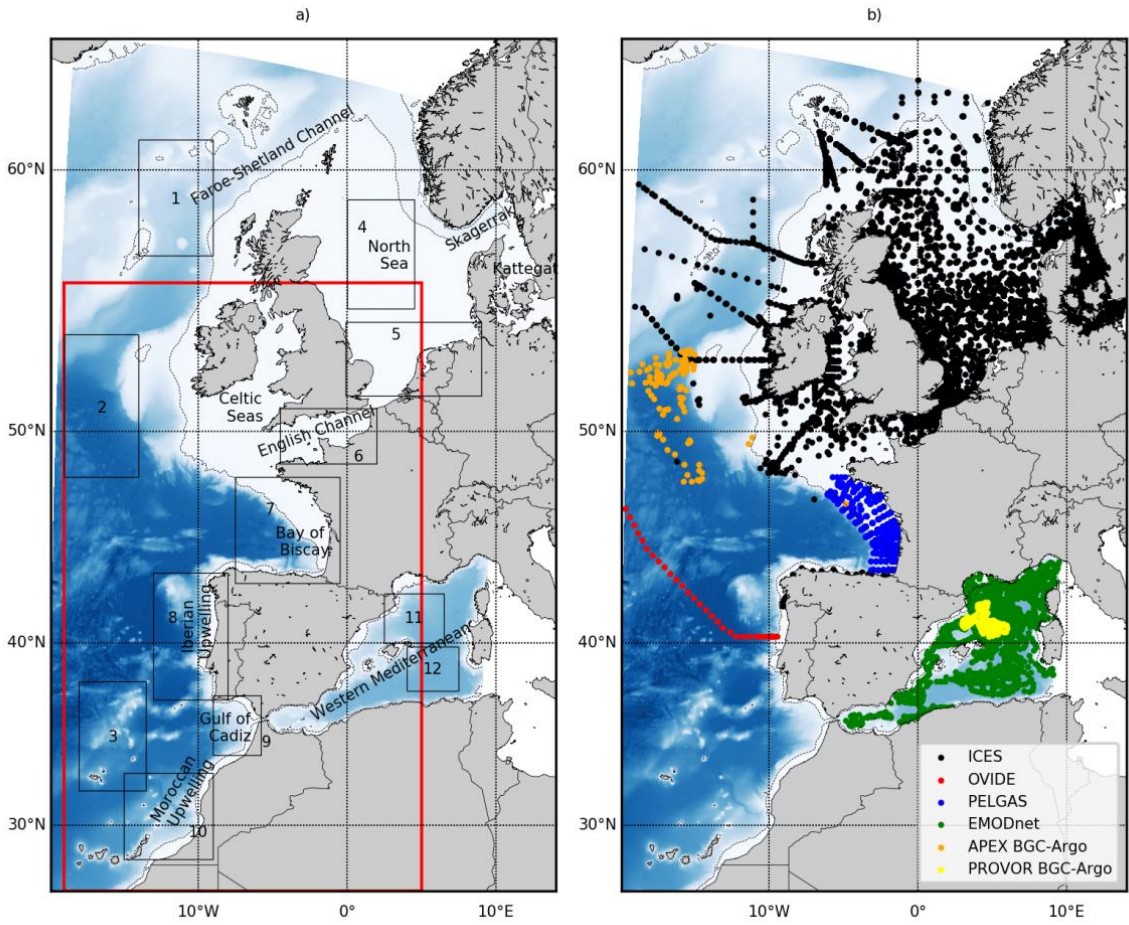

Figure 1. a) IBI Extended Domain on a curvilinear grid and IBI Service Domain extending from -19°E to 5°E and 26°N to 56°N on a regular grid (red rectangle). The 12 back boxes represent the different areas described in Section 2 and used for evaluation in Section 4. They represent the North Atlantic (boxes 1, 2, 3), North Sea (boxes 4 and 5), English Channel (box 6), Bay of Biscay (box 7), Iberian upwelling (box 8), Gulf of Cadiz (box 9), Moroccan upwelling (box 10) and Western Mediterranean (boxes 11 and 12). b) Location of in-situ biogeochemical data used for validation. ICES data are in black, OVIDE section and PELGAS data of the North-East Atlantic EMODnet dataset are respectively in red and blue, the Mediterranean Sea EMODnet dataset is in green, the APEX BGC-Argo float in the Atlantic is in orange, and PROVOR BGC-Argo float in the Mediterranean is in yellow. The blue colour shading represents the bathymetry and dashed line is the 200 m isobath delimiting the shelf region.

940

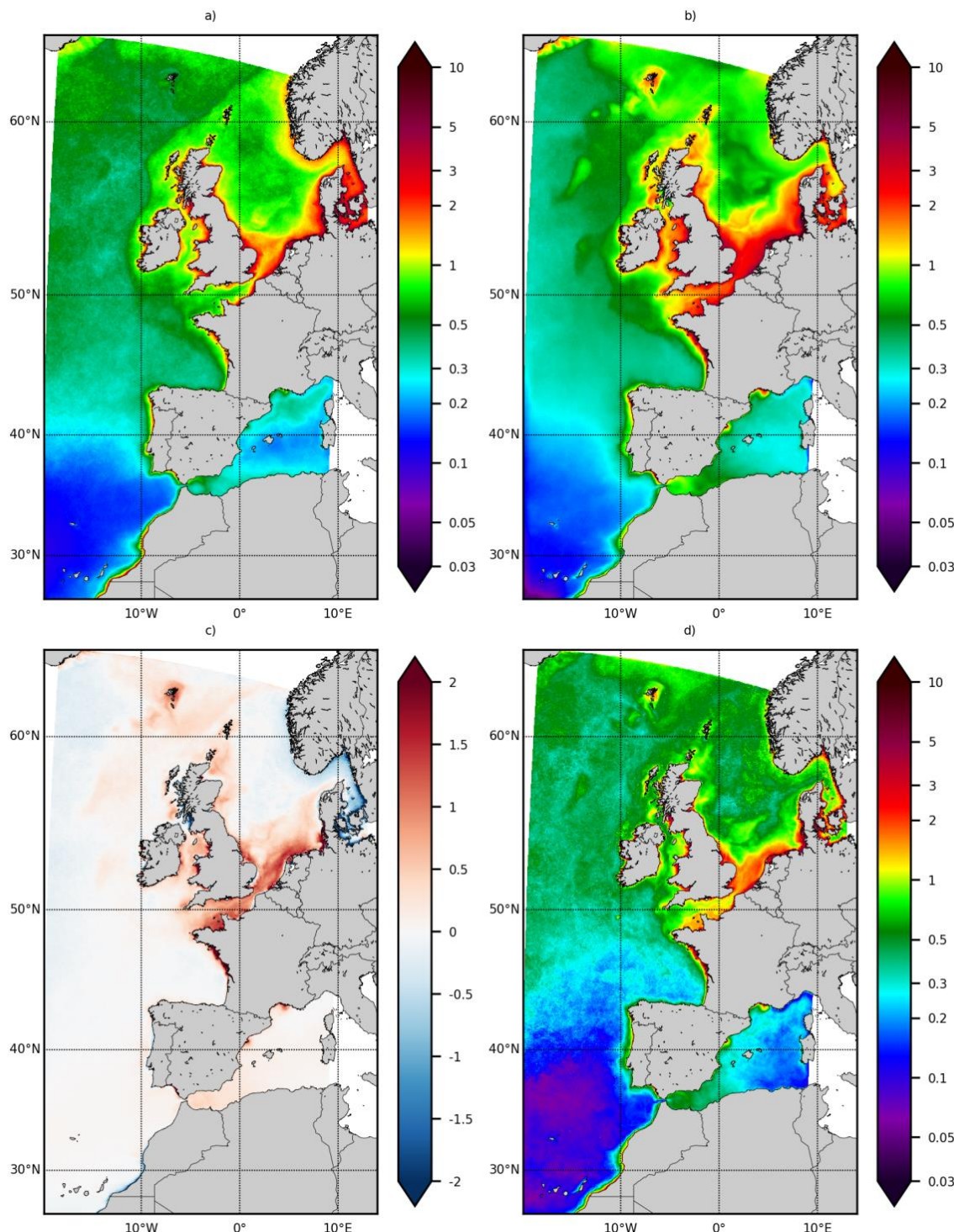

Figure 2. Sea surface Chl-a. a) annual average of ESA OC-CCI ocean colour product, b) annual average of IBI36, c) averaged bias of Chl-a ($model - obs$) and and d) RMSE ( $\sqrt{\langle (model - obs)^2 \rangle}$ ), all expressed in mg Chl m$^{-3}$. Statistics are computed from monthly fields between 2010 and 2016.

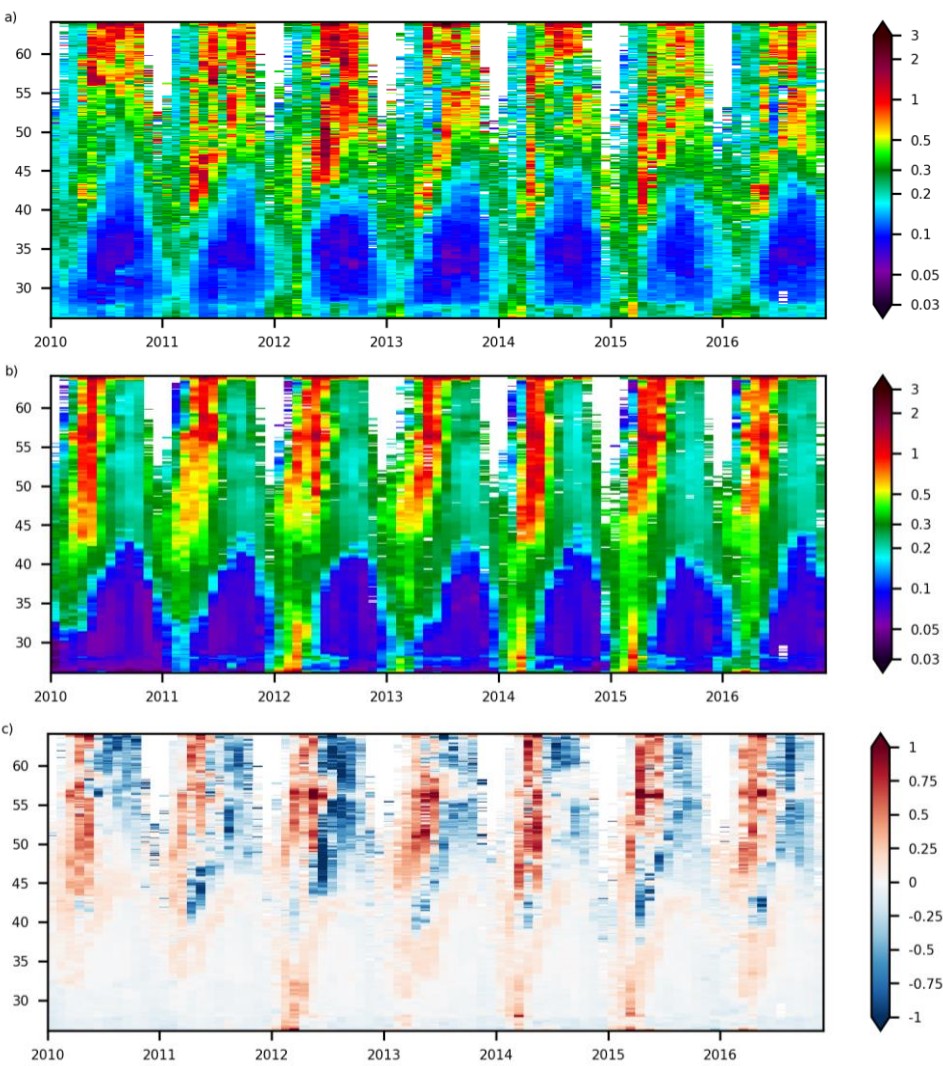

Figure 3. Hovmöller diagram for sea surface Chl-a at 15°W between 2010 and 2016. a) ESA OC-CCI ocean colour product, b) IBI36 and c) bias of Chl-a ($model - obs$), all expressed in mg Chl m$^{-3}$. Monthly fields between 2010 and 2016 are used.

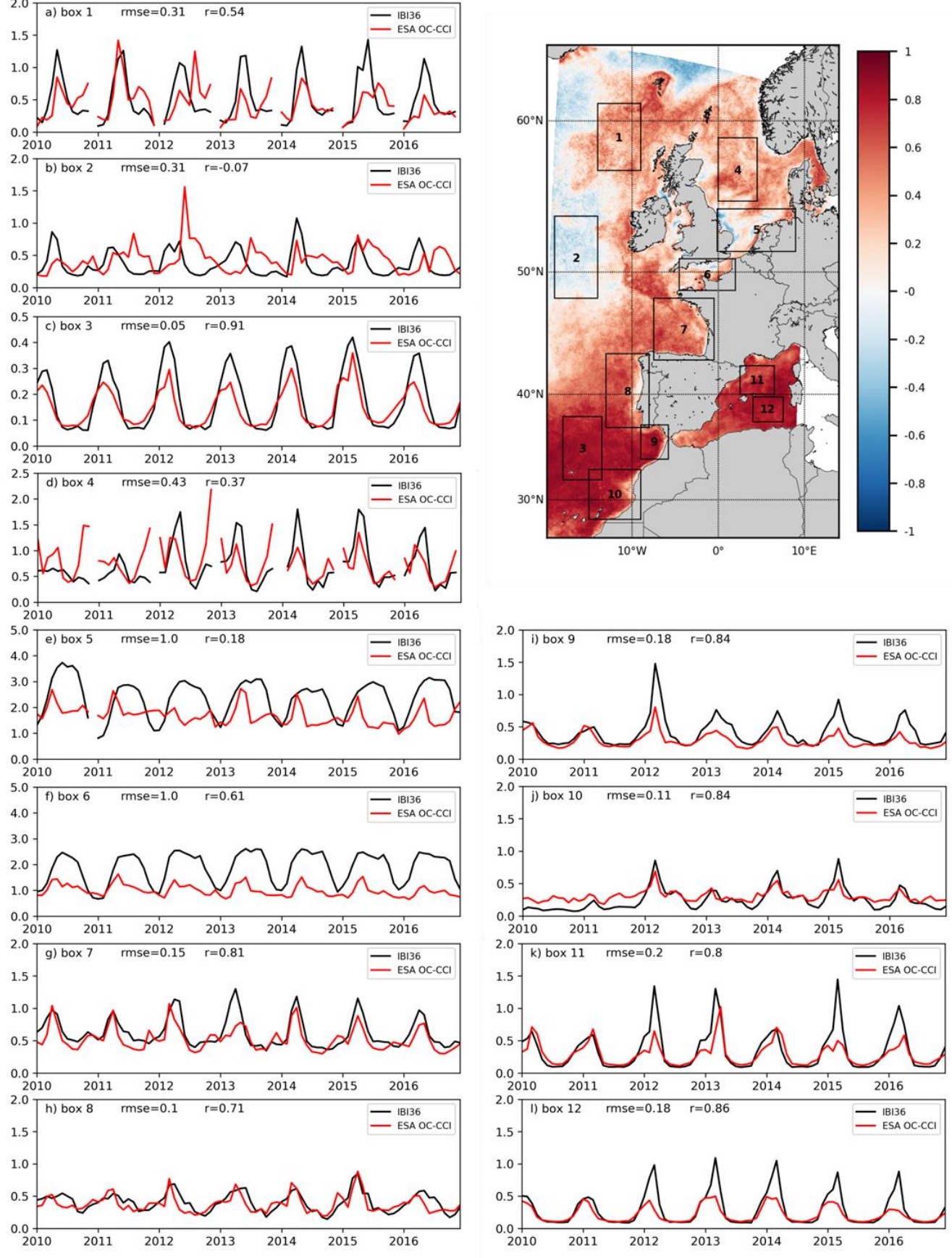

Figure 4. Time series of Sea surface Chl-a (mg Chl m$^{-3}$) between 2010 and 2016. IBI36 is in black and ESA OC-CCI ocean colour product in red. Chl-a is averaged over the 12 boxes defined in Figure 1 and reported to the map on the top-right. The RMSE ( $\sqrt{\langle(model - obs)^2\rangle}$ ) and correlation (r) between the model and the data are indicated for each time series. The top-right panel represents the spatial distribution of temporal correlation between the model and the observation. Note the different scales in y-axis.

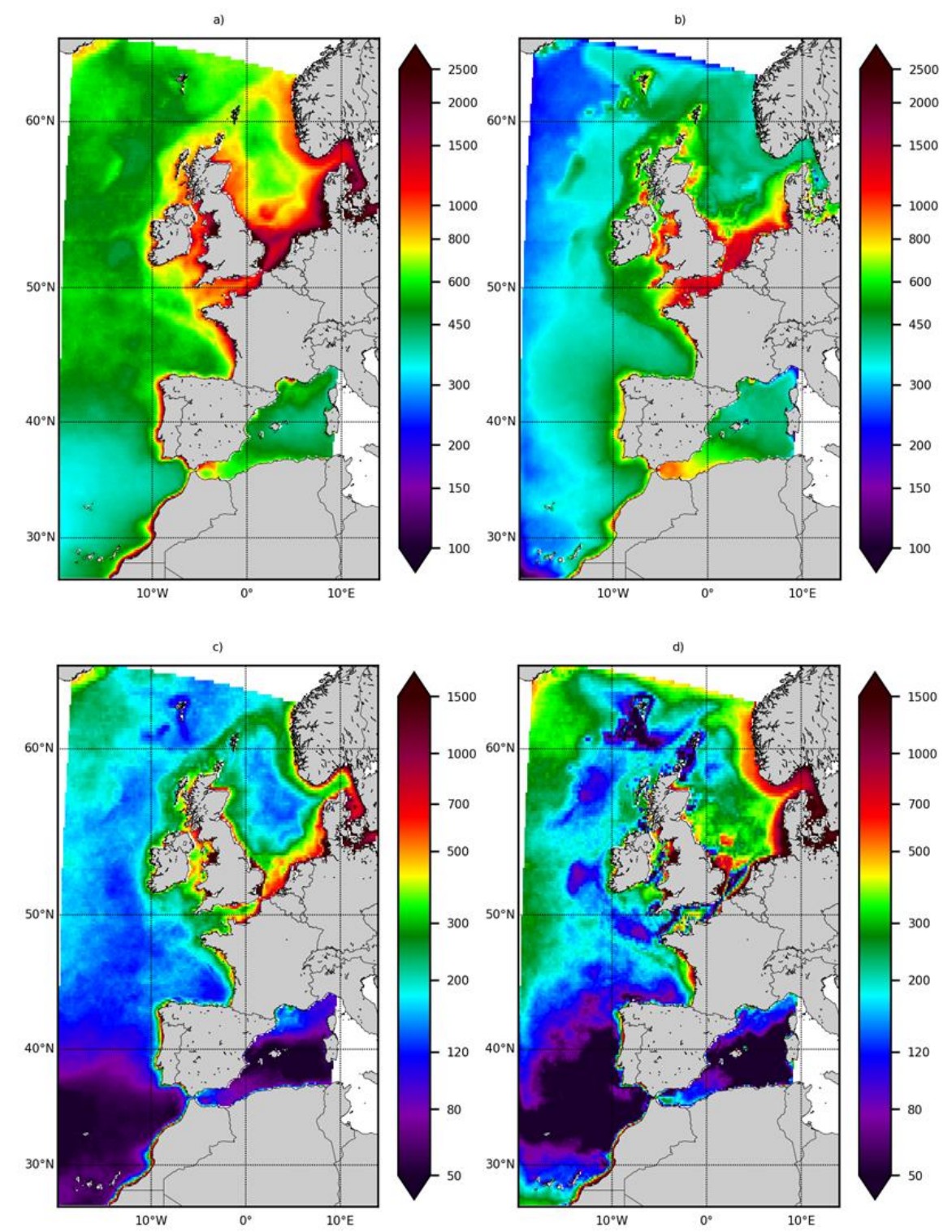

Figure 5. Depth integrated NPP (mg C m$^{-2}$d$^{-1}$). a) Mean of the three NPP products (VGPM, Eppley-VGPM and CbPM), b) IBI36, c) standard deviation of the three NPP products, and d) bias ($IBI36 - mean\ NPP\ poducts$). Statistics are computed from monthly fields between 2010 and 2016.

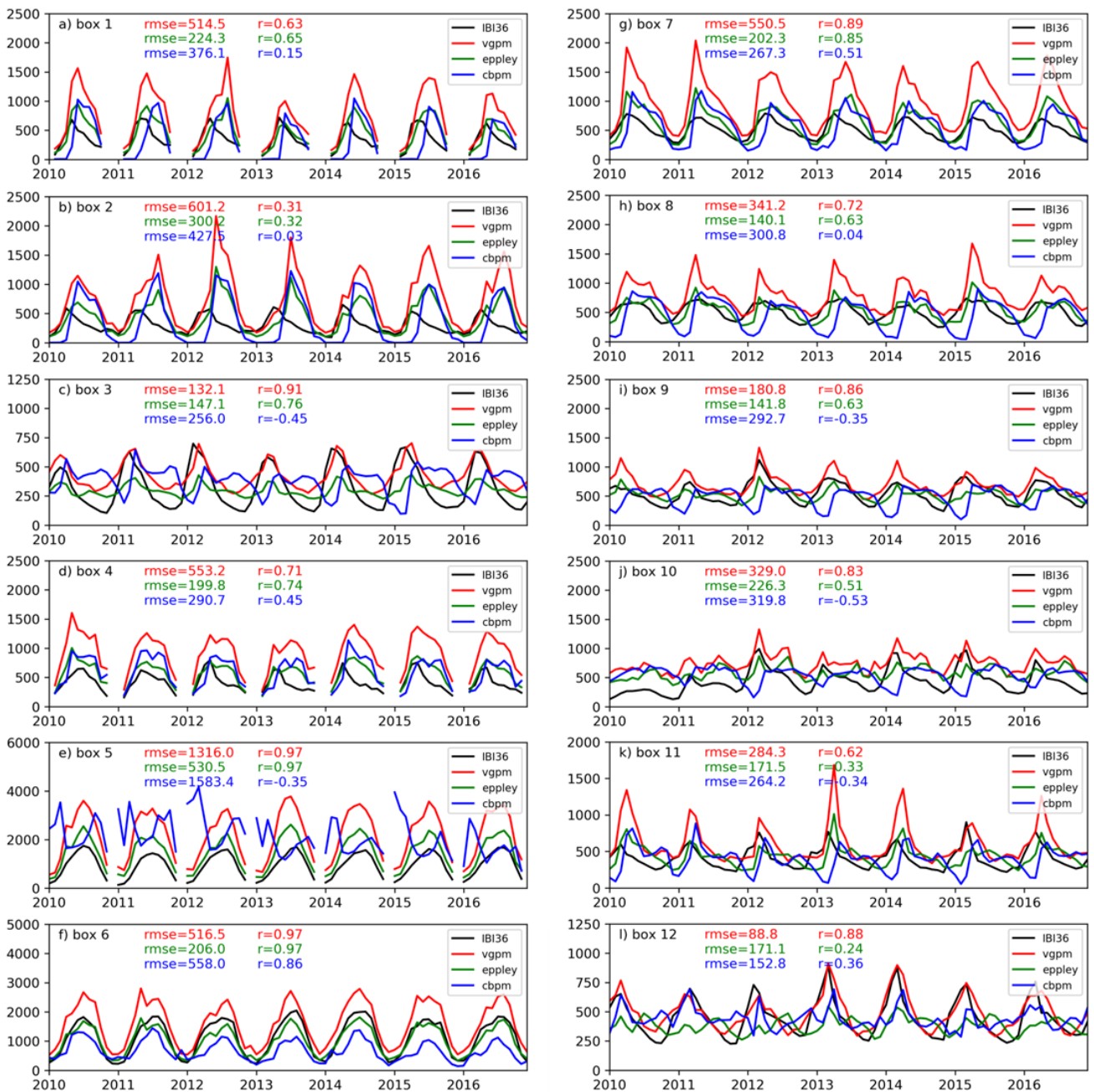

Figure 6. Time series of depth integrated NPP (mg C m$^{-2}$ d$^{-1}$) between 2010 and 2016. IBI36 is in black, VGPM in red, Eppley-VGPM in green and CbPM in blue. NPP is averaged over 12 small boxes defined in Figure 1. Note the different scales in y-axis. The RMSE ( $\sqrt{\langle (model - obs)^2 \rangle}$ ) and correlation (r) between the model and the NPP products (using corresponding colours) are indicated for each time series.


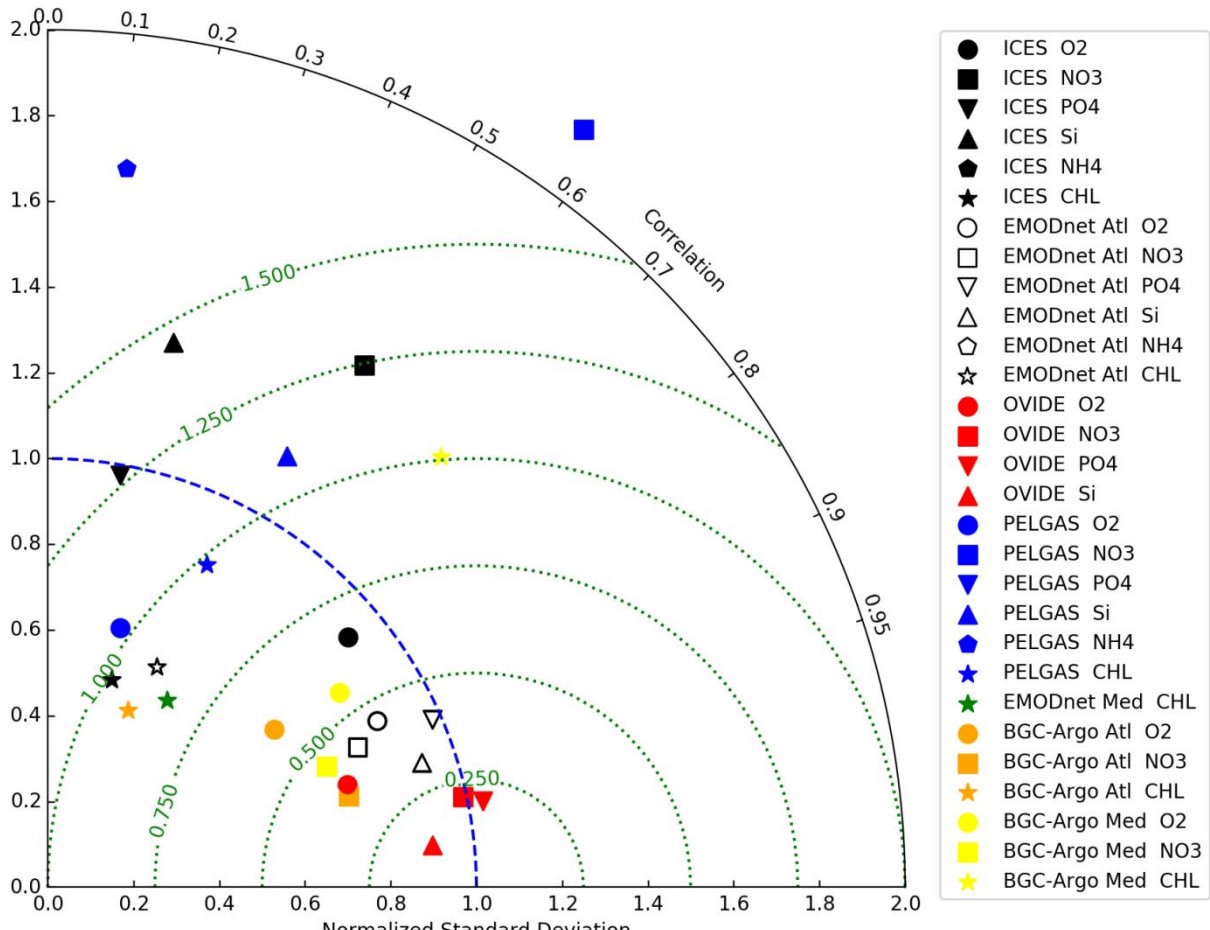

Figure 7. Taylor diagram summarizing the skill of the IBI36 system to estimate the main biogeochemical variables: oxygen (circle), nitrate (square), phosphate (triangle pointing upwards), silicate (triangle pointing down), ammonium (pentagone) and Chl-a (star) from ICES (black), North Atlantic EMODnet product (white), OVIDE (red), PELGAS (blue), Mediterranean EMODnet product (green), APEX BGC-Argo in the Atlantic (orange) and PROVOR BGC-Argo in the Mediterranean (yellow).


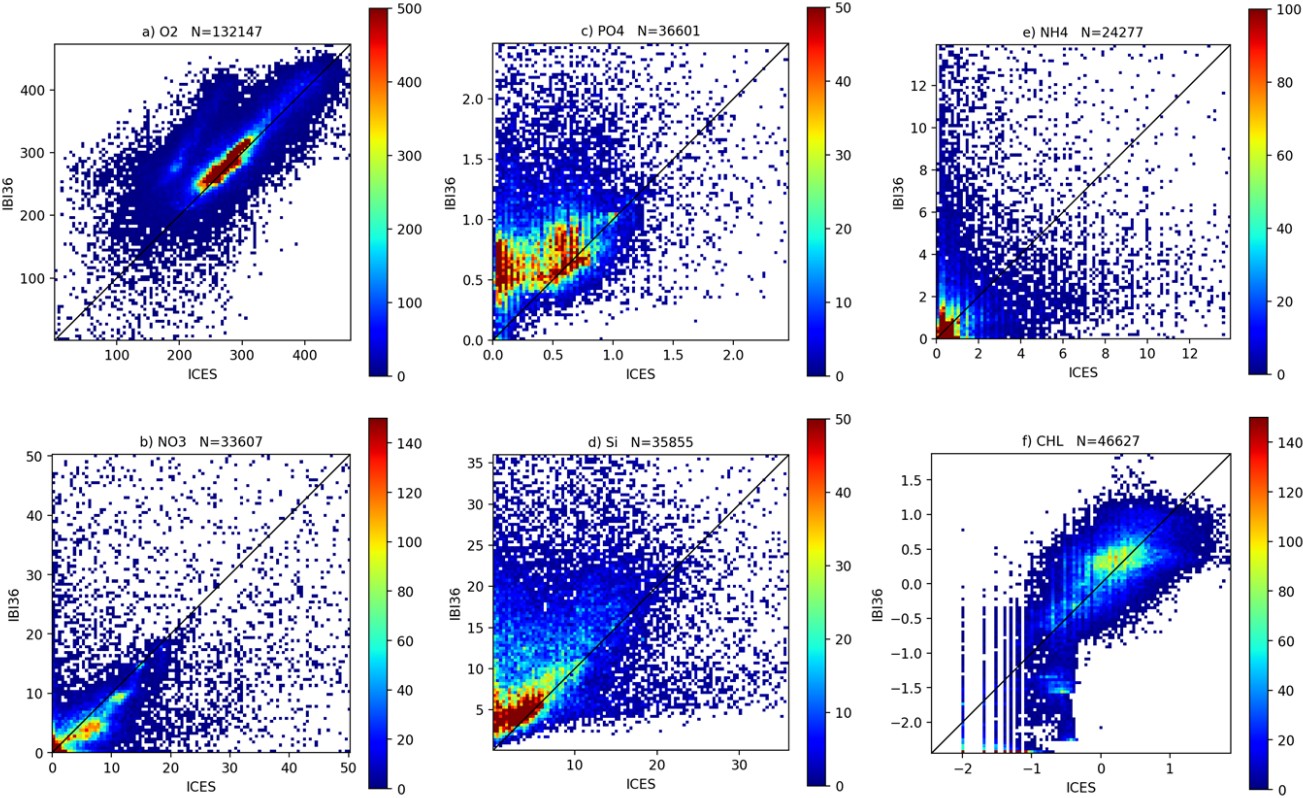

Figure 8. Density plots for a) oxygen, b) nitrate, c) phosphate, d) silicate, e) ammonium, and f) log10(Chl-a). ICES data are on the x-axis and IBI36 on the y-axis. Oxygen and nutrients are expressed in µmol l$^{-1}$. Each axis is divided in 100 bins and colorbar represents the density of the match-ups (number of overlapping points). Note the different scales for the variables. N indicates the total number of match-ups. Daily averaged IBI36 outputs and ICES data are collocated in space and time between 2010 and 2016. All depths are presented, keeping in mind that ICES data are mainly located in the shallow and coastal waters of the Northern seas.

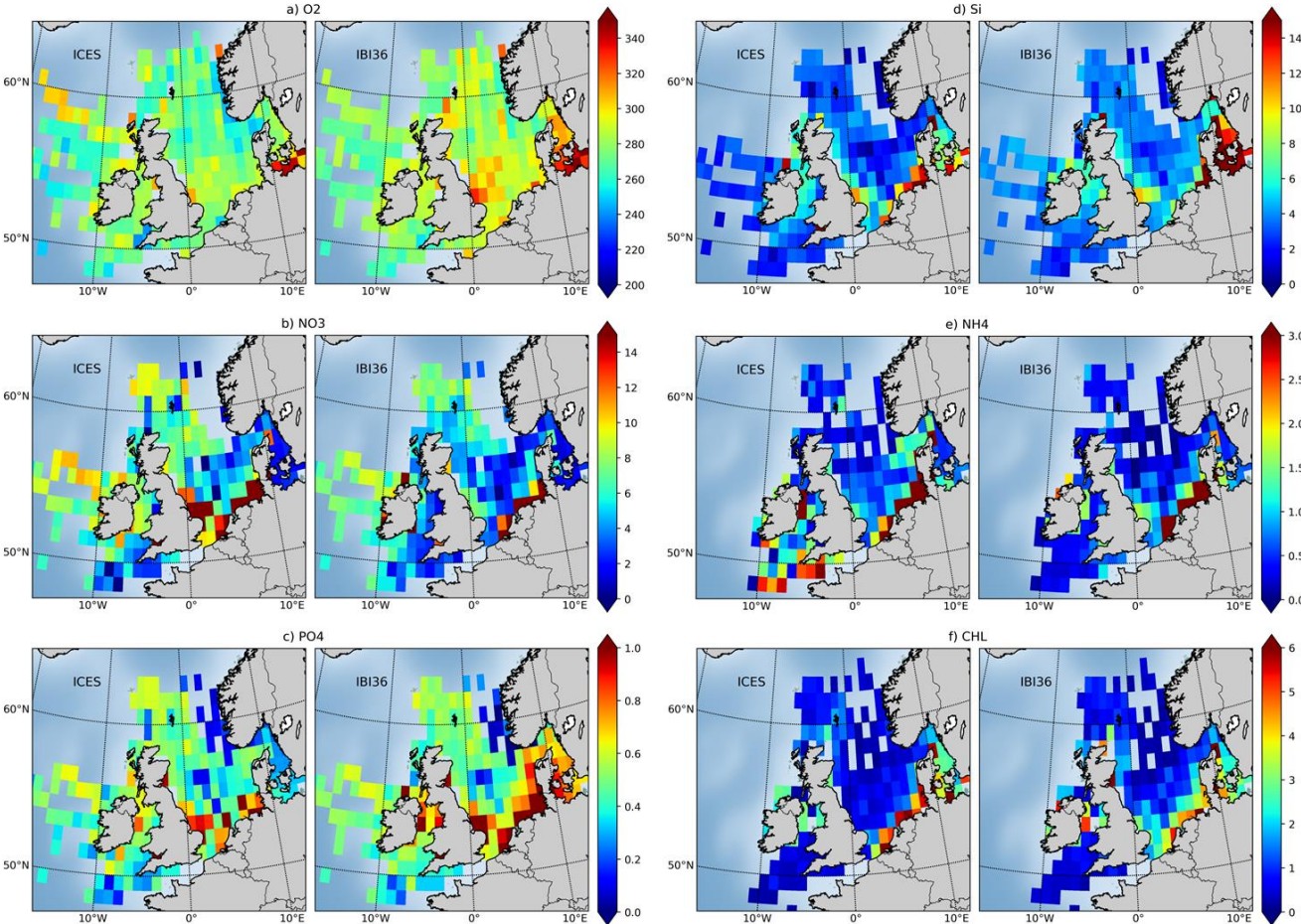

Figure 9. Surface concentrations of a) oxygen, b) nitrate, c) phosphate, d) silicate, e) ammonium and f) Chl-a from ICES database (left of each panel) and IBI36 (right of each panel). Oxygen and nutrients are expressed in µmol l$^{-1}$ and Chl-a in mg Chl m$^{-3}$. Daily averaged IBI36 outputs and ICES data are collocated in space and time between 2010 and 2016. Match-ups are averaged between 0 and 10 meter depth, gridded and averaged on a horizontal grid of 1°x1° resolution.


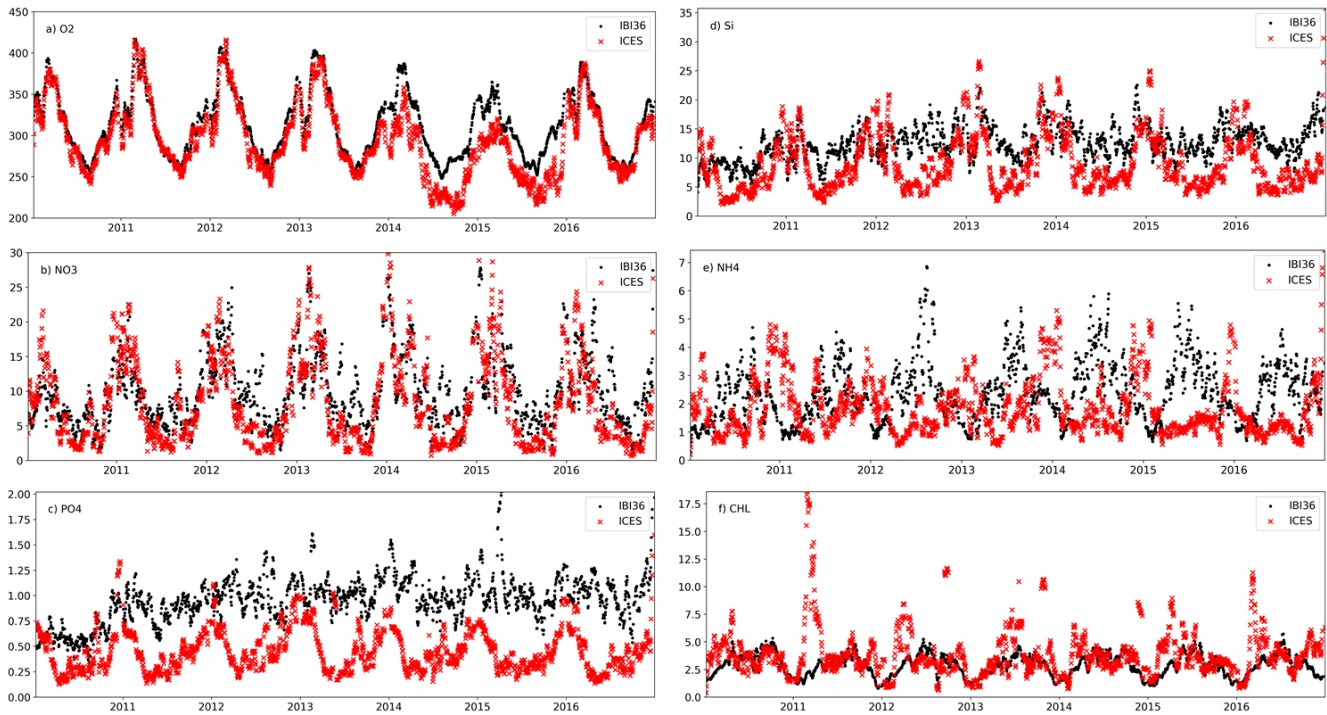

Figure 10. Time series of surface concentrations of a) oxygen, b) nitrate, c) phosphate, d) silicate, e) ammonium and f) Chl-a from ICES (red) and IBI36 (black). Oxygen and nutrients are expressed in µmol l$^{-1}$ and Chl-a in mg Chl m$^{-3}$. Daily averaged IBI36 outputs and ICES data are collocated in space and time between 2010 and 2016. Match-ups are averaged between 0 and 10 meter depth, and daily averaged. Time series are smoothed using a 10-day window.


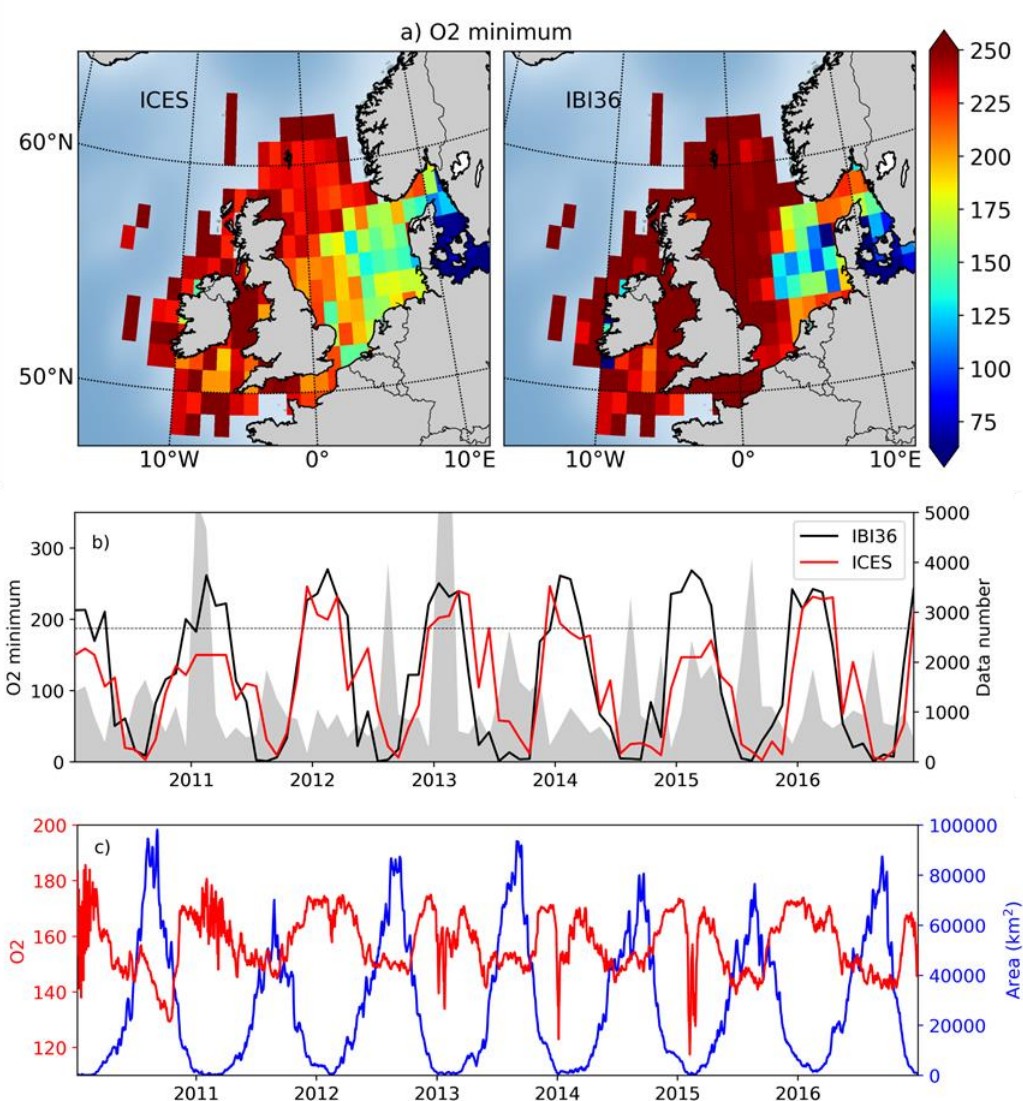

Figure 11: a) Minimum oxygen (µmol l$^{-1}$) in ICES data (left) and IBI36 collocated to ICES (right) between 2010 and 2016. b) Time series of minimum oxygen (µmol l$^{-1}$) in ICES data (red) and IBI36 IBI36 collocated to ICES (black). The deficiency threshold of oxygen (6 mg l$^{-1}$ or 187.5 µmol l$^{-1}$) is represented by the dashed line. Number of available data in ICES is added to the right axis (area plot in gray). c) Surface area (in km$^2$) vulnerable to oxygen deficiency, that is where oxygen decrease below the deficiency threshold (blue; right axis) and associated mean oxygen concentrations (µmol l$^{-1}$; red; left axis) using the whole IBI36 simulation (not only IBI36 collocated to ICES). The three subplots are for the continental shelf (bathymetry <= 200m).

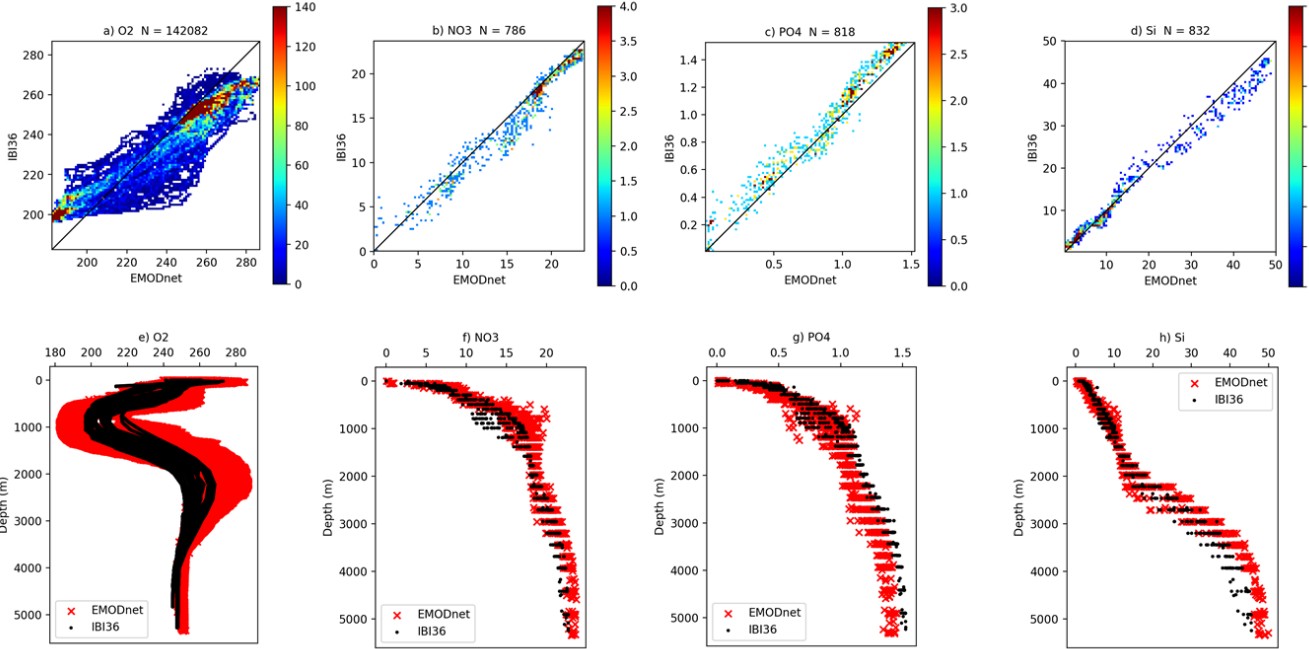

Figure 12. Density plots (top) and vertical profiles (bottom) for oxygen (a, e), nitrate (b, f), phosphate (c, g), and silicate (d, h) from OVIDE section data (EMODnet dataset) and IBI36. All nutrients are expressed in µmol l⁻¹. OVIDE data are on the x-axis and IBI36 on the y-axis of the density plots. Each axis is divided in 100 bins and colorbar represents the density of the match-ups (number of overlapping points). Note the different scales for the variables. N indicates the total number of match-ups. Daily averaged IBI36 outputs and OVIDE section data are collocated in space and time.


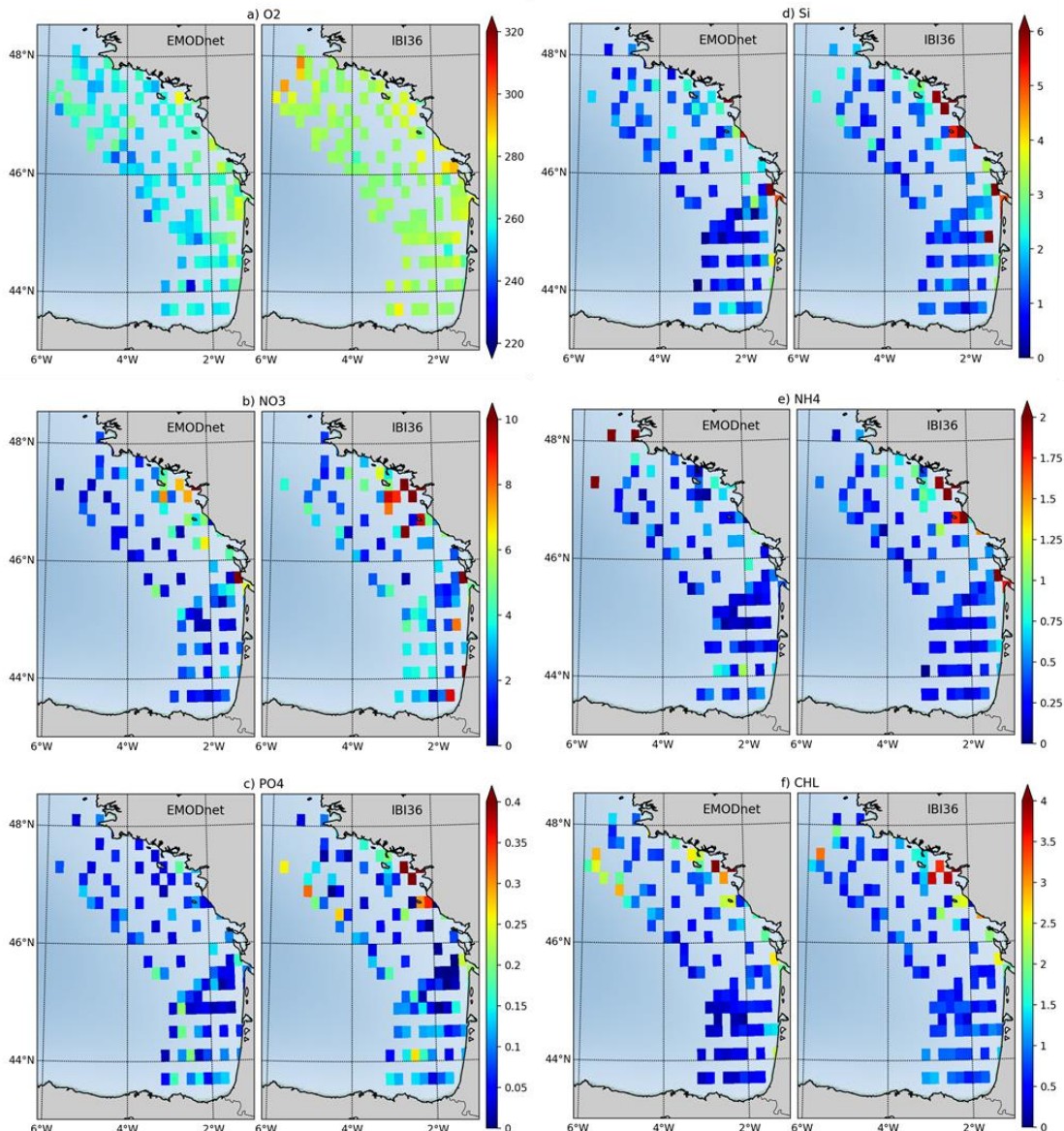

Figure 13. Surface concentrations of oxygen (a), nitrate (b), phosphate (c), silicate (d), ammonium (e) and Chl-a (f) from the PELGAS data of the EMODnet database (left of each panel) and IBI36 (right of each panel). Oxygen and nutrients are expressed in $\mu$mol l$^{-1}$ and Chl-a in mg Chl m$^{-3}$. IBI36 and PELGAS data are collocated in space and time between 2010 and 2016. Match-ups are averaged between 0 and 10 meter depth, gridded and averaged on a horizontal grid of 0.2°x0.2° resolution.

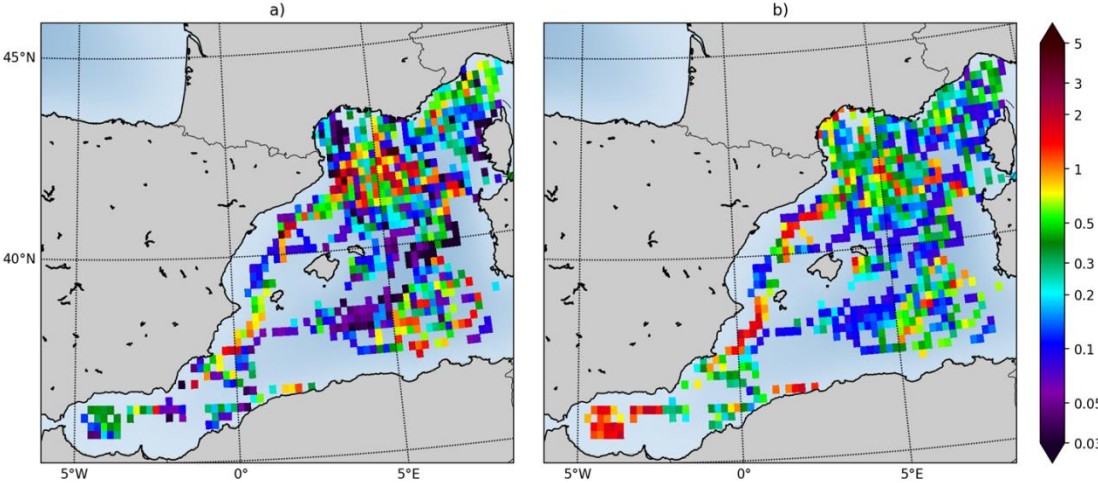

Figure 14. Sea surface Chl-a from EMODnet dataset (a) and IBI36 (b) in mg Chl m$^{-3}$. IBI36 and EMODnet dataset are collocated in space and time between 2010 and 2016. Match-ups are averaged between 0 and 10 meter depth, gridded and averaged on a horizontal grid of 0.2°x0.2° resolution.

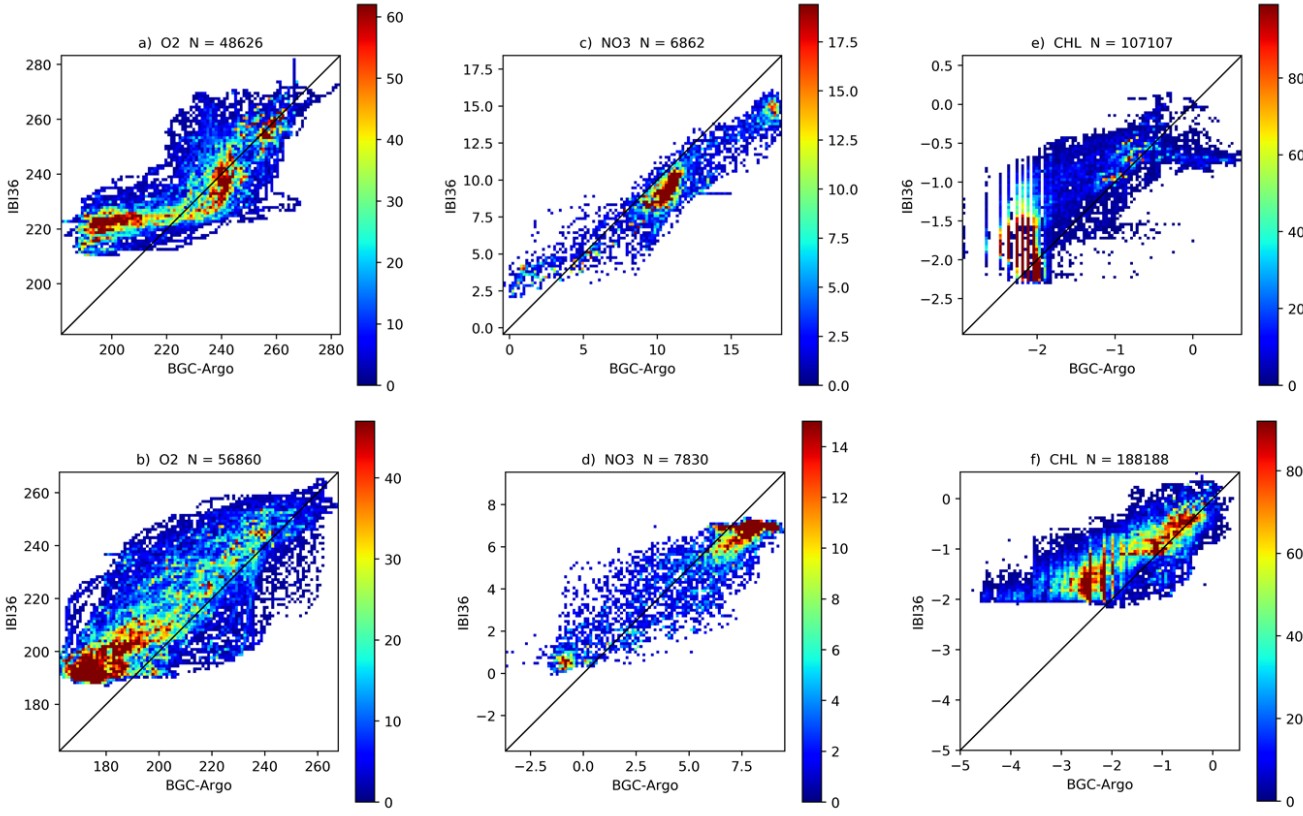


Figure 15. Density plots for oxygen (a, b), nitrate (c, d), and log(Chl-a) (e, f) from BGC-Argo data and IBI36 for the Atlantic (top) and Mediterranean (bottom). Oxygen and nitrate are expressed in µmol l$^{-1}$. Argo data are on the x-axis and IBI36 on the y-axis of the density plots. Each axis is divided in 100 bins and colorbar represents the density of the match-ups (number of overlapping points). Note the different scales for the variables. N indicates the total number of match-ups. IBI36 and Argo

data are collocated in space and time.

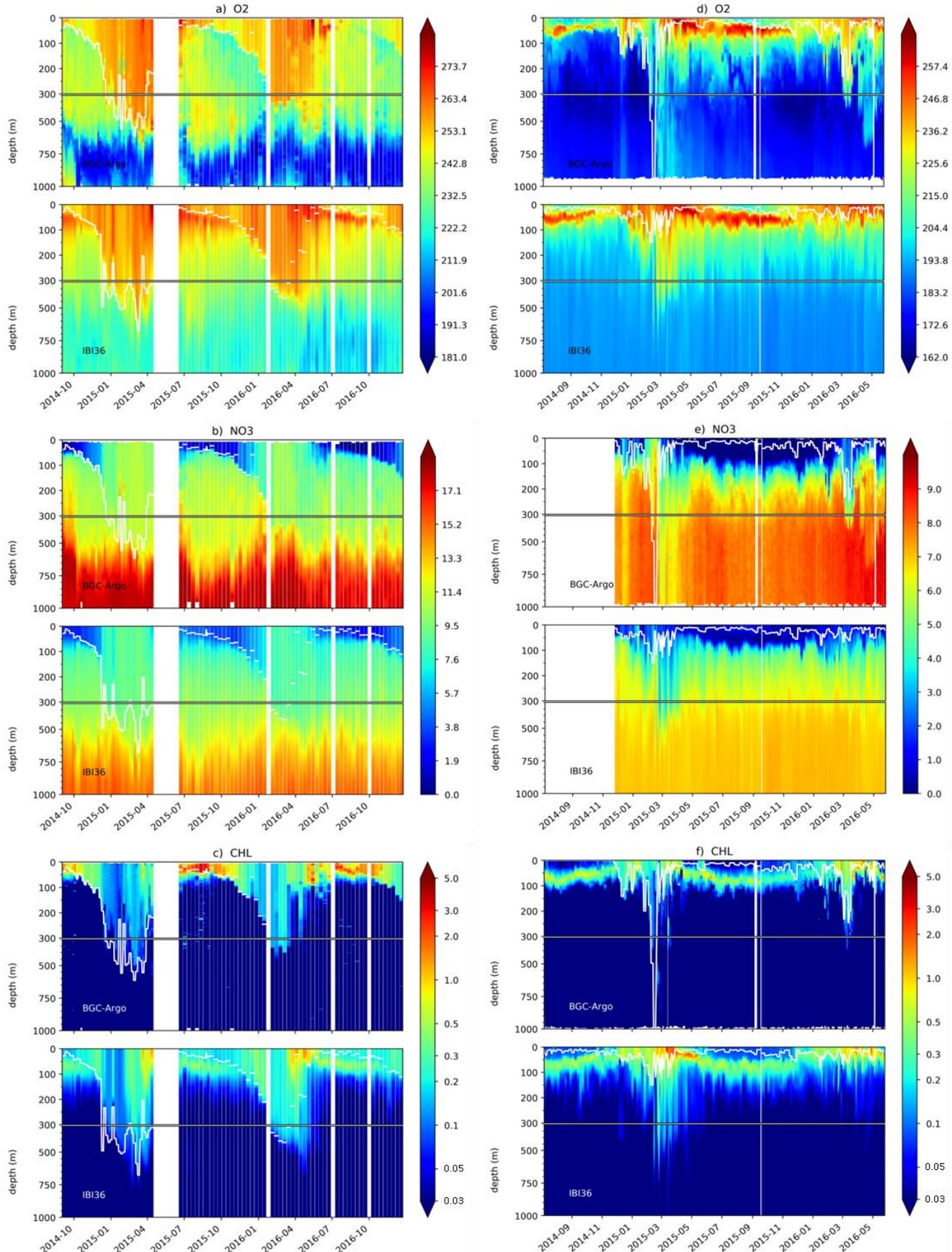

Figure 16. Time series of oxygen (top), nitrate (middle), and Chl-a (bottom) from BGC-Argo data and IBI36 for the Atlantic (left) and Mediterranean (right). Oxygen and nitrate are expressed in µmol l$^{-1}$ and Chl-a in mg Chl m$^{-3}$. IBI36 and Argo data are collocated in space and time. The white line represents the MLD computed from density criterion of 0.03 kg m$^{-3}$ difference from surface.