# Peer review of "Modelling the marine ecosystem of IBI European waters for CMEMS operational applications"

_Ocean Science, 2018_

## Referee Comment (RC1) · Anonymous Referee #1 · 19 Feb 2019

General comments:

The manuscript "A CMEMS forecasting system for the marine ecosystem of IBI" by Elodie Gutknecht et al. aims to describe the skill performances of the CMEMS operational model system "IBI36" for the Iberia-Biscay-Ireland (IBI) area, with specific emphasis on the biogeochemical component. IBI36 is built on a physical-biogeochemical online coupling between NEMOv3.6 physical model and PISCESv2 biogeochemical model, at $1/36°$ horizontal resolution and with 50 vertical levels. The system is operational since April 2018, and consistently provides 7-days forecasts for ocean physics and biogeochemistry. The validation is applied over the "IBI Extended Domain", and is performed with a 2010-2016 simulation, using a suite of different reference data streams (from satellite and in situ, and 2 BGC-Argo floats), and then on a regional

basis with comparison with in situ historical data (Northern Seas, North-East Atlantic waters, Bay of Biscay and Mediterranean Sea). IBI36 results generally consistent with the main properties of biogeochemical variables (chlorophyll, nutrients, oxygen and net primary production), in terms of spatial distribution and seasonal cycles. An indicator concerning oxygen deficiency is also proposed and assessed, which may be of great importance for the broad environmental communities, extending to the general public as well.

The manuscript is clear and well written, with a precise subdivision of the different sections (model configuration, reference data and validation, discussion). However, some major weaknesses can be highlighted:

1. The title specifically refers to the "CMEMS forecasting system": even if it may be acknowledged the goal of the work, the validation is here performed at seasonal or annual time scale on the period 2010-2016, computing monthly averages (as defined in P8, L29), and there is no reference to the forecasting skill of the operational system, that should be done on daily (or at least, weekly) averages. So, how the described validation may be suitable for the short-term forecasting system assessment? The reader would be interested in see results on how the IBI36 short-term forecast products compare with reference data (or with the background simulation). Which metrics, of the ones proposed, can then be used operationally to quantitatively evaluate the forecasting skill?

2. The model system products are validated basically following consistency (Figs. 2 to 6, 8, 9, 10ab, 11, 12, 13, 15), providing BIAS and correlation. However, the uncertainty of products (i.e., forecast, see the previous point) seems not addressed. It may be interesting to quantitatively assess the capability of the model to represent the observed variability: did the author consider to evaluate the model uncertainty, e.g. by means of RMSE? The use of quantitative comparison may be even more useful in the NPP analysis, to estimate the spatial gradient. Further, is there any reason why Fig. 7 does not report the linear regression (as done in Figs. 11 and 14)? Moreover, on a higher

level, given that the validation assessment is discussed following a regional subdivision (which is basically driven by the availability of data), I think it could be useful to provide the readers with a synthesis table with the mean values of the validation metrics (i.e. BIAS, correlation, uncertainty – e.g. as RMSE) per each region.

3. The synthetic overview of the "IBI waters" (Section 2) in terms of the biogeochemical and ecosystem dynamics is interesting since gathers different regions with different properties in a single framework. However, it is not effectively used to define the areas then adopted in the validation (P9, L19). Authors should refer the definition of the 12 small boxes to this overview and then link the different model performances to the different biogeochemical properties of the specific areas. As an example, the availability of reference data may allow validating the consistency of the IBI36 system (or of a specific product) in a specific area/season. And this may be critical to provide indications on the quality of the model system to simulate eutrophication in the different areas (and again, the use of the synthesis table suggested at point 2 would be beneficial for this purpose).

Specific comments:

1. Are there other model applications to the IBI waters (also not operational)? Can the authors discuss how does the IBI36 perform in comparison with other operational systems (e.g. CMEMS Global)? A link with the global model is included only in Section 5 (P15, L15), where some more details (and relevant references) could be added. Also, the comment to the global performance at the eastern boundary (Skagerrak and Kattegat, P12, L16) could be more supported by some reference (again, the CMEMS Global).

2. GODAE Class metrics are widely recognized as references for the skill performance assessment of ocean operational systems and adopted also within the Copernicus community. Is there a specific reason why the authors did not include any mention to them?

3. Do you have any explanation of why the model anticipates the spring-summer bloom (development and decrease) in the north Atlantic (P9, L14-18)? Is it linked with physical forcing or with the biogeochemical parameterization?

4. Fig. 4 and comment at (P9, L19-24): referring to "high correlation coefficient between the model and the data" I suppose the authors consider correlation values (r) larger than 0.7. However, box 12 (Gulf of Cadiz) has a correlation r=0.55, which is actually lower than the one for the English Channel (box 6, r=0.59). Please comment. Further in Fig. 4: how the associated error to ESA OC-CCI product is estimated?

5. Which specific products have been used from EMODnet? Please clarify the kind of "regional aggregated products" used (P7, L31-33). Further, you refer to EMODnet database as "in situ" (P10, L23; L13, L10, and P14, L25), but this looks in contradiction with the "aggregation" previously referred. Please clarify and reformulate.

6. Fig. 10 is potentially very interesting, especially for communities more linked to environmental management. I suggest to enrich the discussion and provide some more information on this indicator: how is computed the minimum in Fig.10a and b (is this defined considering at least 1 daily value lower than the threshold as in P11, L17? is the minimum found along the vertical profile, or at the sea bottom? That should be defined at the beginning of the discussion at P11)? Do the two maps show the absolute minimum over the whole investigated period for each cell shown? What is shown in the time series of Fig.10c? Please clarify. Further, the deficiency threshold (at 187 umol/l) could be added to Fig.10c (e.g. with a dashed line): this may help to appreciate the model skill in representing the oxygen deficiency, and the fact that in 2011 and 2015 the model predicts higher levels than observed. To measure the skill of the model to overcome a specific threshold in comparison with observed data, authors may also consider using the statistics based on the relative operating characteristic (ROC), as for example shown in Sheng and Kim (J. Mar. Sys. 76, 212-243, 2009).

7. Figs. 14 and 15 show the comparison with BGC-Argo floats data in Atlantic and West
Mediterranean areas covered by the IBI system. May the authors add any comment on the systematic behavior of the model in comparison with the float also for nitrate and oxygen (as done for Chl-a at depth, see P14, L2, and then in the discussion, P15, L13)? Do the authors suspect that this behavior is related to possible errors of the model (due to representativeness or biogeochemical parameterization), or may be also related to uncertainty associated to float sensors? Further, does the white line in Fig. 15 represent the MLD? If so, it should be added to the caption, and briefly explained how it was computed.

8. In the discussion (P15, L1-6) why do the authors write that the higher biases on the continental shelf are "no surprising"? Is this related to larger uncertainty given by the river input? Or is there any limits due to the physical drivers? If it seems obvious, please clarify. Secondly, is there any literature basis for the NPP (e.g. with in situ measurements) that can be used to establish which of the 3 referred products we should trust more?

9. At the end of the discussion, there is a reference to OMI (P16, L8-11): this might not be interesting for the general reader. However, since there is a specific indication of the "IBI-MFC biogeochemical forecast service", this should be properly validated before providing OMI (see major point n.1).

Technical/other corrections:

In the webpage of OSD (https://www.ocean-sci-discuss.net/os-2018-161/#discussion), the correct author's surname should be "Sotillo" and not "Sottilo".

P2, L8 and P4, L31. The "IBI-MFC Team" has not been defined.

P3, L14-15. "The annual primary production is then limited its seasonal variations are limited": not clear, please correct or reformulate.

P3, L29. References about N:P ratio around 20 in the Western Mediterranean are also found in the recent work of Lazzari et al. (Deep-Sea Res. I 108, 39-52, 2016; see their

[Figure]

Fig. 7), that may be added.

P5, L25-27. "Although PISCES was originally designed for global ocean applications, the distinction of two phytoplankton size classes and the description of multiple nutrient co-limitations allow the model to represent ocean productivity and biogeochemical cycles in the major ocean biogeographic provinces (Longhurst, 1998)." The reference seems not appropriate: PISCES is not used in Longhurst (1998), which only refers to the bio-provinces. Please reformulate.

P6, L1-2. "To respect the conservation of the tracers, the coupling between biogeochemical and physical components is done every other time." Not clear, please clarify: what do the authors consider as "conservation of the tracers"? Is it mass conservation? This question also relates to the reference to Aumont et al. (2015), at L19-L34: it would be clearer for the reader to add more details about the river inputs (which are very important in coastal regions). Please describe the specific inputs (also reporting from Aumont, Section 4.9.2, and the one from EEA), list the names of the rivers (or their total amount), specify what you consider as "natural" and what as "anthropic", and describe their impact in the investigated region. Finally, please clarify the use of "reminder" the last sentence "For the other variables, a reminder of the initial conditions is given.": does this mean the initial condition values are maintained constant during the simulation?

P6, L8. "is" should be "are".

P7, L20. As done with VGPM and CbPM, please add a reference also to the "Eppley version".

P8, L6-18. Could you provide which fields from the BGC-Argo repository have been used and whether additional calibrations were performed after?

P8, L18. "Johnson et al." (dot missing).

P11, L13. "Breitburg et al." (dot missing).

P11, L19. It should be "surface area in winter".

P11, L20. The authors refer to "the west coast of France" but in Fig.10 a and b it looks like only the north-western French coast is covered by data, and with value larger than 200 umol/l. Please check.

P12, L4. "The seasonal cycle is in phase for Chl-a but out of phase for ammonium" it is also possible to refer to Fig.7e, where correlation for NH4 is r=0.2.

P14, L25. Instead of "is described here", it should be clearer to indicate the corresponding Section or Figure (Figs. 2, 4 for Chl-a and 5, 6 for NPP). Or are the authors here referring to the general approach of the manuscript? Please clarify. Further, it should be "are described".

P17, L3. The fact that a co-author is also acknowledged sounds somehow strange. Please check.

P29, L3. Correct "Eppley".

P33, Fig. 10. Add the units for O2 minimum (umol/l).

―――――――――――――――

---

## Referee Comment (RC2) · Anonymous Referee #2 · 13 Mar 2019

The main objective of the article "A CMEMS forecasting system for the marine ecosystem of IBI European waters" is to demonstrate the performances of the CMEMS operational 1/36° coupled NEMO/PISCES of the IBI (extended) area from a 2010-2016, 7 years model simulation. The authors state that the model is in relatively good agreement with observations (for the main biogeochemical variables) and reproduce the large scale main biogeochemical processes, with a focus on seasonal cycles. In addition, an interesting oxygen deficiency indicator is presented. The article includes a wide variety of comparisons (15 figures) computed from an impressive set of observation. The document is well written and a very valuable effort has been done to make it as clear as possible. It results in a complete review of PISCES ability to simulate large scale main biogeachemical features in the IBI domain. In my opinion, despite these

undeniable qualities, the article miss some important points and there are several major issues that should be (at least mostly) adressed before publication. The "general recommendations section" describes major points that, in my opinion, need to be improved before publication. The second section only provides some specific comments that could help to address the general recommmandation needs. I did not go into very specific details as i think some materials should be added (and modified) before going further.
* * *
General recommendations

A) In the actual form, the article is rather a review (interesting) of results from a 7 years PISCES simulation in the IBI region. It not clear to me whether the objectives are 1) demonstrate the ability of this system for regional forecast applications or 2) to assess PISCES ability to represent the main biogeochemical features of the IBI region => suitable for operational applications. In any case, important modifications need to be made and the introduction should be extended to better mention the objectives and the means used to adress these objectives. From the sentence p2, line27, i guess the main objective is more on point 2) but the way it is written gives the impression that the authors want to assess the relevancy of this configuration for operational 7 day forecasts which is clearly not at all demonstrated here (ex 4.1.1. first sentence "Predicted sea surface..." should be rather something like "the model sea surface chlorophyll..." ). This comment also underlines a lack of focus. The result section is more a presentation of various diagnosis than a scientific analysis of the results which might be induced by too unclear objectives of the paper. I understand that the paper is more an analysis of "consistency" but the reader can sometimes be irritated when a figure is nearly not discussed. If you decide to mainly make this scientific analysis in the discussion part i would suggest to dedicate a full section to this discussion (instead of discussion and conclusion) and to clearly state this point in introduction. Finally, taking into account a focus on objective 2), we sometimes lose the purpose of the paper and i am not sure

that the article really answer whether or not this PISCES simulation is well adapted to operational simulation in the IBI area (i am sure it is). In my opinion this it mainly the consequence of an efficient but also too straight way of writting with a lack of methodology : "the role of this figure is to show this point, which demonstrates this information, which is related to my main objectives in this way". Nevertheless, i am sure that this could be corrected quite easily as there is also a very robust amount of information.

B) One major issue is the total absence of informations regarding uncertainties (except in figure 4). In my opinion, this point is crucial to assess the performance of any simulation even if the authors decide to only focus on point 2). As this might be difficult (and not necessary) to consider uncertainty for every comparison, i suggest to add one section dedicated to uncertainties. It should both discuss uncertainties of the PISCES simulation and the observation products. One simple question that should be addressed: is the model simulation included within the range of observation uncertainty ? For instance, first order uncertainties could be deduced from the statistical level of dispersion (in a specific radius representative of error correlations). This particular suggestion might probably be better adapted for comparison with ocean colour data (or when a large amount of data is available and could be presented with histograms).

C) Particularly using a 1/36° model resolution, it is quite a shame not to show simple weekly or monthly mean maps (at least for chlorophyll) highlighting this PISCES simulation ability to catch the variability existing in ocean colour data. This would be relevant to insit on the benefits of using a 1/36° resolution. Are there such 1/36 PISCES simulation elsewhere ? What are the benefits compare to lower resolution models ? Why do you need this resolution here ? I really think that it would be of great importance to compare your results with other existing configurations or models. This would also be a great help to demonstrate that your configuration is well adapted to simulate the IBI biogeochemical features (to solve some of the general recommendation A) ). If it is really not possible to perform comparions this have to be clearly stated in introduction. More generally, the introduction completely miss the state of the art part. This definitely

has to be corrected.

D) In order to make the document easier to read (in particular considering the probable adding figures (from my previous comments), i also suggest to add an annexe section. Indeed some figures are only quickly and partially discussed (for instance : first paragraph of section 4,2,1 only 8 lines to described 3 figures ; in section 4,1,1, 4 figures are covered ; fig 7,8,9 f) are not discussed... ). This would also allow to better focus the article on its objectives. (This comment is connected with the general recommandation 1) )

\*\*\*\*\*\*\*\*\*\*\*\*\*\*\*\*\*\*\*\*\*\*\*\*\*\*\*\*\*\*\*\*\*\*\*\*\*\*\*\*\*\*\*\*\*\*\*\*\*\*\*\*\*\*\*\*

Main Specific comments

Considering the major recommendations, i here only specify, by section, the main specific comments as there will probably have a second review process. Generally, i do think that some accuracy is needed.

1) Introduction

- See section major recommendations. It has to be extended in order to clearly explain the objectives of the article and present a clear state of the art (in terms of existing forecast studies with PISCES and on models on the IBI region) in the area.

2) IBI European waters Interesting and quite complete part. It is a nevertheless a shame that the link between this part and the result section are nearly non-existent.

3) The IBI36 configuration - What is the influence of a $1/4°$ (bio) and $1/12°$ (physics) initialisation in the $1/36°$ simulation ? Do you have an idea how long this information is kept in the system, about the differences ? Don't you think it can strongly impact the first timing and intensity of the blooms (especially with an initialization in January ? Why don't you make a few years spin-off ? - I would be very interested to have some informations on differences between a $1/4°$ and a $1/36°$ simulations. It would also help to justify the use of your configuration. Some additional information about the nutrient

forcing files would be welcome as i suppose that it could mainly explain most of the the coastal deviations with observations. Do you have an idea of probable impacts of using $1/2°$ data to $1/36°$ grid, how do you deal with this ?

4) IBI36 evaluation 4.1 Satellite estimations

- How is the temporal correlation figure 2d calculated ? From monthly averages ? - On which grid are the differences calculated (model or verification) ? Are there impacts resulting from this grid changing ? In particularly for Net primary production comparison (1/6 degree compared to 1/36 degree). Please clarify this point as non linearities can have significant effects. - It is difficult to see something in figure 3. The discussion on the bloom timing could also be done using different boxes of figure 4. This would permit to remove one figure (or in annex). - p6. lines 18-21 you say that Net primary production estimates are model products ? If it is true why do you include these data sets in a section 4,1 called satellite estimations ?

4,2 In situ historical data

- In fig 8, for oxygen, it would be clearer if you could modify the colorbar. At a first view we think that the data and the model are very different while the bias is only 4% - It is a shame you do not discuss at all some possible reasons why the model does not catch the low oxygen period in 2014-2015. Especially when you thereafter discuss about oxygen deficiencies... -Don 't you think that one of the main limitation comes from the nutrient forcing files ? Could you specify a little bit more (than it is in 3.2) as it seems to be quite important ? Where are the anthropic inputs located in the model ? What is the impact of these additional anthropic inputs ? It could be relevant to go a little bit deeper into this point.

4,3 Argo data

How are the Argo data co-located with the model data ? Do you grid Argo data on the model grid ? Could you re-precise dates ? Although correlations are still hight, results

are much less good than previously. Do you have an idea why ? Small scale features ? Have you compared the Argo data with some of the previous observations ? (it also could help in defining uncertainty levels)

4,4 Discussion and conclusion

- I would suggest to separate the discussion and the conclusion since the authors have clearly decided not to deeply analyze the results in the result section. The discussion proposed here is interesting but should be extended and also make a clear link better with the objectives that should be first clarified in introduction.
* * *

---

## Author Comment (AC1) · 14 Jun 2019

Dear Referee,

Please find our comments/responses in blue throughout your text.
We have also attached a new release of the article.
This is a major revision of the initial article: the introduction has been completely revised, the sections are modified, a discussion section has been added. A synthesis of the results has been added by using a table and a Taylor diagram.
All your comments have been taken into account in the new release of the article.
If required we can also provide a version with the word track change, but I don't think that's helpful to you.
Thank you for your useful comments.

Elodie Gutknecht on behalf of co-authors.

**General comments:**

The manuscript "A CMEMS forecasting system for the marine ecosystem of IBI" by Elodie Gutknecht et al. aims to describe the skill performances of the CMEMS operational model system "IBI36" for the Iberia-Biscay-Ireland (IBI) area, with specific emphasis on the biogeochemical component. IBI36 is built on a physical-biogeochemical online coupling between NEMOv3.6 physical model and PISCESv2 biogeochemical model, at 1/36° horizontal resolution and with 50 vertical levels. The system is operational since April 2018, and consistently provides 7-days forecasts for ocean physics and biogeochemistry. The validation is applied over the "IBI Extended Domain", and is performed with a 2010-2016 simulation, using a suite of different reference data streams (from satellite and in situ, and 2 BGC-Argo floats), and then on a regional basis with comparison with in situ historical data (Northern Seas, North-East Atlantic waters, Bay of Biscay and Mediterranean Sea). IBI36 results generally consistent with the main properties of biogeochemical variables (chlorophyll, nutrients, oxygen and net primary production), in terms of spatial distribution and seasonal cycles. An indicator concerning oxygen deficiency is also proposed and assessed, which may be of great importance for the broad environmental communities, extending to the general public as well.

The manuscript is clear and well written, with a precise subdivision of the different sections (model configuration, reference data and validation, discussion). However, some major weaknesses can be highlighted:

1. The title specifically refers to the "CMEMS forecasting system": even if it may be acknowledged the goal of the work, the validation is here performed at seasonal or annual time scale on the period 2010-2016, computing monthly averages (as defined in P8, L29), and there is no reference to the forecasting skill of the operational system, that should be done on daily (or at least, weekly) averages. So, how the described validation may be suitable for the short-term forecasting system assessment? The reader would be interested in see results on how the IBI36 short-term forecast products compare with reference data (or with the background simulation). Which metrics, of the ones proposed, can then be used operationally to quantitatively evaluate the forecasting skill?

**Author answer:**
You are right; the title explicitly refers to the "CMEMS forecasting system". I realize that the title was not judiciously chosen and does not reflect the purpose of this paper. The purpose of this paper is to evaluate the biogeochemical component of the IBI coupled model system. This system has been developed to monitor and forecast the ocean dynamics and marine ecosystems of the European waters. But prior to its operational launch, a pre-operational qualification simulation is necessary to evaluate the performance of the coupled system. Here we evaluate this qualification simulation. It represents a substantial paper, with many figures. But it's necessary and voluntary. This paper represents the first validation of the biogeochemical component of the IBI36 system; it is intended to be complete and detailed, as it will serve as a basis for further studies. So the forecasting capability of the operational system will be studied and described in another paper. To avoid any misunderstanding, we propose to modify the title of the paper by: "Modelling the marine ecosystem of IBI European waters for CMEMS operational applications". We also modified the abstract and the Introduction to make clearer the objective of the paper.

2. The model system products are validated basically following consistency (Figs. 2 to 6, 8, 9, 10ab, 11, 12, 13, 15), providing BIAS and correlation. However, the uncertainty of products (i.e., forecast, see the previous point) seems not addressed. It may be interesting to quantitatively assess the capability of the model to represent the observed variability: did the author consider to evaluate the model uncertainty, e.g. by means of RMSE? The use of quantitative comparison may be even more useful in the NPP analysis, to estimate the spatial gradient. Further, is there any reason why Fig. 7 does not report the linear regression (as done in Figs. 11 and 14)? Moreover, on a higher level, given that the validation assessment is discussed following a regional subdivision (which is basically driven by the availability of data), I think it could be useful to provide the readers with a synthesis table with the mean values of the validation metrics (i.e. BIAS, correlation, uncertainty – e.g. as RMSE) per each region.

**Author answer:**
You are right, in order to complete the skill assessment, the RMSE was added in the Chl-a comparison to satellite estimate. The NPP analysis was also revised. It now presents the mean of

the three NPP products (VGPM, Eppley-VGPM and CbPM), the standard deviation of these three products and the bias between the IBI36 system and the averaged NPP products.

The regression lines are not discussed, so they are removed in all figures.

We also added a synthesis to summarize the performances for each region. It is presented in a Table for comparison to satellite estimates and in a Taylor diagram for in-situ observations.

3. The synthetic overview of the "IBI waters" (Section 2) in terms of the biogeochemical and ecosystem dynamics is interesting since gathers different regions with different properties in a single framework. However, it is not effectively used to define the areas then adopted in the validation (P9, L19). Authors should refer the definition of the 12 small boxes to this overview and then link the different model performances to the different biogeochemical properties of the specific areas. As an example, the availability of reference data may allow validating the consistency of the IBI36 system (or of a specific product) in a specific area/season. And this may be critical to provide indications on the quality of the model system to simulate eutrophication in the different areas (and again, the use of the synthesis table suggested at point 2 would be beneficial for this purpose).

**Author answer:**
You are right, the "IBI waters" (Section 2) has been revised in order to introduce the areas (the 12 small boxes) adopted in the validation section. Model performances are now better linked to the different biogeochemical properties of the specific areas.

**Specific comments:**

1. Are there other model applications to the IBI waters (also not operational)? Can the authors discuss how does the IBI36 perform in comparison with other operational systems (e.g. CMEMS Global)?

**Author answer:**

Within the framework of CMEMS, three other MFC share a part of their domain with IBI:
- GLO-MFC which covers the world's oceans at 1/4° resolution and is also using the PISCES biogeochemical model,
- MED-MFC which covers the Mediterranean Sea at 1/24° with the Biogeochemical Flux Model (BFM; Vichi et al., 2007a,b),
- NWS-MFC which covers the North-West European Shelf at 1/15° latitudinal resolution and 1/9° longitudinal resolution (~ 7km) with the ERSEM ecosystem model (Baretta et al., 1995).

For the physical component, two intercomparison papers have been submitted in ocean Science - Special issue "The Copernicus Marine Environment Monitoring Service (CMEMS): scientific advances". They are Lorente et al. (2019) and Mason et al. (2019).

For the biogeochemical component, the comparison of the different model applications is a work in progress and will be the subject of a separate paper, including the contribution of the regional in relation to the global, as well as the comparison of three distinct biogeochemical models.

The Introduction has been revised and now includes a state of the art part.

**References:**

Baretta, J. W., W. Ebenhoh, et al. (1995). "The European regional seas ecosystem model, a complex marine ecosystem model." Netherlands Journal of Sea Research 33(3-4): 233-246.
Lorente, P., García-Sotillo, M., Amo-Baladrón, A., Aznar, R., Levier, B., Sánchez-Garrido, J. C., Sammartino, S., De Pascual, Á., Reffray, G., Toledano, C., and Álvarez-Fanjul, E.: Skill assessment of global, regional and coastal circulation forecast models: evaluating the benefits of dynamical downscaling in IBI surface waters, Ocean Sci. Discuss., https://doi.org/10.5194/os-2018-168, in review, 2019.
Mason, E., Ruiz, S., Bourdalle-Badie, R., Reffray, G., Garcia-Sotillo, M., and Pascual, A.: Copernicus (CMEMS) operational model intercomparison in the western Mediterranean Sea: Insights from an eddy tracker, Ocean Sci. Discuss., https://doi.org/10.5194/os-2018-169, in review, 2019.
Vichi, M., Pinardi, N., and Masina, S.: A generalized model of pelagic biogeochemistry for the global ocean ecosystem. Part I: theory, J. Marine Syst., 64, 89-109, https://doi.org/10.1016/j.jmarsys.2006.03.006, 2007a.
Vichi, M., Masina, S., and Navarra, A.: A generalized model of pelagic biogeochemistry for the global ocean ecosystem. Part II: numerical simulations, J. Marine Syst., 64, 110-134, https://doi.org/10.1016/j.jmarsys.2006.03.014, 2007b.

A link with the global model is included only in Section 5 (P15, L15), where some more details (and relevant references) could be added. Also, the comment to the global performance at the eastern boundary (Skagerrak and Kattegat, P12, L16) could be more supported by some reference (again, the CMEMS Global).

**Author answer:**
The link with the global model is made in Section 3.2 as it is used at initial and open boundary conditions of the IBI system. The comment to the global performance at the eastern boundary (Skagerrak and Kattegat) has been modified. There is not particular evaluation of the Global system is this region. The CMEMS Baltic Sea regional configuration instead of the global product should be tested at the eastern open boundary of the IBI36 system.

2. GODAE Class metrics are widely recognized as references for the skill performance assessment of ocean operational systems and adopted also within the Copernicus community. Is there a specific reason why the authors did not include any mention to them?

**Author answer:**
You are right; it's an oversight on our part. GODAE metrics are introduced in Section 4.

3. Do you have any explanation of why the model anticipates the spring-summer bloom development and decrease in the north Atlantic (P9, L14-18)? Is it linked with physical forcing or with the biogeochemical parameterization?

**Author answer:**

The model anticipates the spring-summer bloom development and decrease in the north Atlantic. The bloom onset is in phase in the south part of the domain, but spreads more rapidly to the north as compared to satellite data. The summer decrease after the bloom is then earlier and sometimes more pronounced in the model. In fact, modelled chlorophyll-a is found deeper during summer with the formation of a Deep Chlorophyll Maximum while maximum Chl-a remains at the surface in BGC-Argo estimates.

I think that both phenomena (timing of the bloom and vertical migration of the maximum chlorophyll) are linked. As phytoplankton starts growing earlier in the model, the mixed layer becomes nutrient-depleted at the end of spring (oligotrophic conditions) in the model and phytoplankton migrates just below. This behavior is also present in the global model.

Several experiments on the physical and biogeochemical components are being carried out. The analysis of the BGC-Argo opens new doors for us to understand the vertical dynamics of phytoplankton. Another area for improvement is opened up with the assimilation of water color data.

This problem is now discussed in the Discussion section.

4. Fig. 4 and comment at (P9, L19-24): referring to "high correlation coefficient between the model and the data" I suppose the authors consider correlation values (r) larger than 0.7. However, box 12 (Gulf of Cadiz) has a correlation r=0.55, which is actually lower than the one for the English Channel (box 6, r=0.59). Please comment.

**Author answer:**
High correlation coefficient refers to values (r) larger than 0.7.
Boxes were slightly revised. As this box was not discussed, it has been removed in the revised version.

Further in Fig. 4: how the associated error to ESA OC-CCI product is estimated?

**Author answer:**
ESA OC-CCI distributes an estimation of the error associated to the chlorophyll-a concentration in sea water. The name of this variable is "CHL_error". But the term "error" is misleading. This is actually a standard deviation of the daily data when calculating the monthly average. It has to be analyzed jointly with the number of daily files that contained data incorporated into the monthly average. So to avoid any misunderstanding, it has been removed in the revised version.

5. Which specific products have been used from EMODnet? Please clarify the kind of "regional aggregated products" used (P7, L31-33). Further, you refer to EMODnet database as "in situ" (P10, L23; P13, L10, and P14, L25), but this looks in contradiction with the "aggregation" previously referred. Please clarify and reformulate.
* * *
**Author answer:**
In the two projects Seadatanet and EMODnet chimie, regional datasets are developed. They are "aggregated datasets" or "observation collections" by region, including one product in the North Atlantic and one in the Mediterranean basin. These datasets receive additional quality control of metadata and data.
They can be found here:
 http://www.emodnet-chemistry.eu/products/catalogue#/search?fast=index&_content_type=json&from=1&to=20&sortBy=popularity&any=aggregated%20datasets

EMODnet website explicitly uses the term "aggregated dataset", and not "aggregated product" as referred in the paper. So to avoid any misunderstanding, the term "aggregated product" has been changed to "aggregated dataset" in the revised paper.
* * *
6. Fig. 10 is potentially very interesting, especially for communities more linked to environmental management. I suggest to enrich the discussion and provide some more information on this indicator: how is computed the minimum in Fig.10a and b (is this defined considering at least 1 daily value lower than the threshold as in P11, L17? is the minimum found along the vertical profile, or at the sea bottom? That should be defined at the beginning of the discussion at P11)? Do the two maps show the absolute minimum over the whole investigated period for each cell shown? What is shown in the time series of Fig.10c? Please clarify. Further, the deficiency threshold (at 187 umol/l) could be added to Fig.10c (e.g. with a dashed line): this may help to appreciate the model skill in representing the oxygen deficiency, and the fact that in 2011 and 2015 the model predicts higher levels than observed. To measure the skill of the model to overcome a specific threshold in comparison with observed data, authors may also consider using the statistics based on the relative operating characteristic (ROC), as for example shown in Sheng and Kim (J. Mar. Sys. 76, 212-243, 2009).
* * *
**Author answer:**
Fig. 10a present the absolute minimum oxygen concentrations for each 1°x1° grid point from ICES dataset (Fig. 10a left) and from IBI36 daily outputs co-located to ICES dataset (Fig. 10a right).
Fig. 10b presents the time series of the absolute minimum oxygen concentrations for each date from ICES dataset (red line) and from IBI36 daily outputs co-located to ICES dataset (black line).
So, in Fig. 10a, and b, we only catch low oxygen concentrations reported to the ICES database. We miss all other low oxygen events. But the objective is here to evaluate/validate the model estimates. The model performs in reproducing the spatial distribution and seasonal evolution of low oxygen concentrations.

After this positive evaluation, we are confident in the model results, so we can estimate the total surface area vulnerable to oxygen deficiency over the continental shelf (bathymetry <= 200m). In Fig. 10c, we do not restrict to the IBI36 daily outputs co-located to ICES dataset. We do not use ICES dataset anymore. But instead we use the full outputs of the simulation to give estimation over the whole IBI domain. We consider that a model grid point is an area vulnerable to oxygen deficiency if at least one daily value decrease below the threshold of 6 mg l$^{-1}$ (or 187.5 µmol l$^{-1}$) during the simulation length, as in Ciavatta et al. (2016). The minimum is found along the vertical profile over the continental shelf (bathymetry <= 200m).

More information or clearer information is now given to describe Fig. 10 in the revised paper. The deficiency threshold is now added to Fig.10b as a dashed line.

The model predicts higher levels than observed in 2011 and 2015 (Fig. 10b), but they come from a few measurement points very close to the coast in the vicinity of river mouth, not captured by the IBI36 system.

The ROC (relative operating characteristic) score is a very interesting skill assessment method to measure the skill of the model to overcome a specific threshold in comparison with observed data. I didn't know this metrics, and I think the biogeochemistry modelling community is not used to seeing this metrics. It therefore needs to be presented and explained in detail, as in Sheng and Kim (J. Mar. Sys. 76, 212-243, 2009). This would add complexity to the paper and this section. But I keep this metrics in mind carefully for a study dedicated to oxygen vulnerable areas.

7. Figs. 14 and 15 show the comparison with BGC-Argo floats data in Atlantic and West Mediterranean areas covered by the IBI system. May the authors add any comment on the systematic behavior of the model in comparison with the float also for nitrate and oxygen (as done for Chl-a at depth, see P14, L2, and then in the discussion, P15, L13)? Do the authors suspect that this behavior is related to possible errors of the model (due to representativeness or biogeochemical parameterization), or may be also related to uncertainty associated to float sensors? Further, does the white line in Fig. 15 represent the MLD? If so, it should be added to the caption, and briefly explained how it was computed.

**Author answer:**

Low oxygen concentrations are systematically overestimated and high nitrate are systematically underestimated. This systematic behavior of the model in comparison with the float is also observed when compared to other observational datasets. It can be related to possible errors of the model (due to representativeness, physical or biogeochemical parameterization).

But uncertainty associated to float sensors can not be excluded. An important work on the quality control is currently being done by the BGC-Argo teams, and the quality check process is evolving very rapidly. Temporal drifts, constant or even non-constant vertical bias, and negative concentrations (see the nitrate in Fig. 15d) are still observed in the BGC-Argo data.

Nitrate and oxygen are now more discussed in the revised paper.

Yes, the white line in Fig. 15 represents the MLD. It is now added to the caption, and its computation is now explained.

8. In the discussion (P15, L1-6) why do the authors write that the higher biases on the continental shelf are "no surprising"? Is this related to larger uncertainty given by the river input? Or is there any limits due to the physical drivers? If it seems obvious, please clarify. Secondly, is there any literature basis for the NPP (e.g. with in situ measurements) that can be used to establish which of the 3 referred products we should trust more?

**Author answer:**

Continental margins are very productive regions and play an important role in the biogeochemical cycle of nutrients and carbon. They are the site of complex interactions between physical, chemical and biological processes that include exchanges between shelf and the open ocean, sediment-water interactions, air-sea fluxes, and land-ocean freshwater inputs. In addition, coastal systems are locally strongly affected by human activities. All these interactions make the continental shelf a challenge to obtain realistic models.
It is now clarified in the revised paper.

Regarding the NPP, we are currently collecting the estimates reported in the literature in order to build an in-situ database. This work is very time-consuming, and unfortunately cannot be included in this paper.

9. At the end of the discussion, there is a reference to OMI (P16, L8-11): this might not be interesting for the general reader. However, since there is a specific indication of the "IBI-MFC biogeochemical forecast service", this should be properly validated before providing OMI (see major point n.1).

**Author answer:**
The text has been modified to avoid confusing the reader.

**Technical/other corrections:**

In the webpage of OSD (https://www.ocean-sci-discuss.net/os-2018-161/#discussion), the correct author's surname should be "Sotillo" and not "Sottilo".

The author's surname has been corrected.

P2, L8 and P4, L31. The "IBI-MFC Team" has not been defined.

The "IBI-MFC Team" is now defined in the Introduction.

P3, L14-15. "The annual primary production is then limited its seasonal variations are limited": not clear, please correct or reformulate.

The sentence has been reformulated to : "The annual primary production is then limited and so is its seasonal variations".

P3, L29. References about N:P ratio around 20 in the Western Mediterranean are also found in the recent work of Lazzari et al. (Deep-Sea Res. I 108, 39-52, 2016; see their Fig. 7), that may be added.

The reference Lazzari et al. (2016) is now added.

P5, L25-27. "Although PISCES was originally designed for global ocean applications, the distinction of two phytoplankton size classes and the description of multiple nutrient co-limitations allow the model to represent ocean productivity and biogeochemical cycles in the major ocean biogeographic provinces (Longhurst, 1998)." The reference seems not appropriate: PISCES is not used in Longhurst (1998), which only refers to the bio-provinces. Please reformulate.

The reference Longhurst (1998) is now removed to avoid any misunderstanding.

P6, L1-2. "To respect the conservation of the tracers, the coupling between biogeochemical and physical components is done every other time." Not clear, please clarify: what do the authors consider as "conservation of the tracers"? Is it mass conservation?

Yes, we refer to mass conservation. The sentence has been modified.

This question also relates to the reference to Aumont et al. (2015), at L19-L34: it would be clearer for the reader to add more details about the river inputs (which are very important in coastal regions). Please describe the specific inputs (also reporting from Aumont, Section 4.9.2, and the one from EEA), list the names of the rivers (or their total amount), specify what you consider as "natural" and what as "anthropic", and describe their impact in the investigated region. Finally, please clarify the use of "reminder" the last sentence "For the other variables, a reminder of the initial conditions is given.": does this mean the initial condition values are maintained constant during the simulation?

More details and clearer explanation is now given about the external inputs of nutrients linked to the river runoffs. We briefly describe the impact of additional nutrient inputs in the investigated region. The terms "natural" and what as "anthropic" have been removed to make the text clearer

and avoid any misunderstanding. The location of the 33 rivers considered in the IBI36 system is available in Maraldi et al. (2013).

Yes, the other variables are maintained constant during the simulation at the open boundary conditions that are the river points. This last sentence "For the other variables, a reminder of the initial conditions is given" is now removed. It is not useful because we are only talking about nutrient inputs here.

We are aware that river discharges could be greatly improved. But it is hampered by the lack of in situ observations.

P6, L8. "is" should be "are".

"is" has been changed to "are".

P7, L20. As done with VGPM and CbPM, please add a reference also to the "Eppley version".

A reference is now added in the revised paper.

P8, L6-18. Could you provide which fields from the BGC-Argo repository have been used and whether additional calibrations were performed after?

The APEX float observations in the Atlantic are adjusted following Johnson et al. (2017). The three variables are "delayed mode" data. For the PROVOR float observations in the Mediterranean, oxygen and nitrate are "Real time" data and Chl-a is "adjusted" data. They are adjusted following Mignot et al. (2018).

P8, L18. "Johnson et al." (dot missing).

The dot was added.

P11, L13. "Breitburg et al." (dot missing).

The dot was added.

P11, L19. It should be "surface area in winter".

It is now corrected.

P11, L20. The authors refer to "the west coast of France" but in Fig.10 a and b it looks like only the north-western French coast is covered by data, and with value larger than 200 umol/l. Please check.

As detailed above, Fig. 10a only catches low oxygen concentrations reported to the ICES database. All other low oxygen events are missed. In Fig. 10c, IBI36 outputs over the whole IBI domain are used to estimate the total surface area vulnerable to oxygen deficiency. So areas not detected by the ICES database may appear vulnerable due to simulation. It is now better explained in the text.

P12, L4. "The seasonal cycle is in phase for Chl-a but out of phase for ammonium" it is also possible to refer to Fig.7e, where correlation for NH4 is r=0.2.

The reference to Fig. 7e is added.

P14, L25. Instead of "is described here", it should be clearer to indicate the corresponding Section or Figure (Figs. 2, 4 for Chl-a and 5, 6 for NPP). Or are the authors here referring to the general approach of the manuscript? Please clarify. Further, it should be "are described".

"is described here" has been changed to "are described in this paper".

P17, L3. The fact that a co-author is also acknowledged sounds somehow strange. Please check.

The acknowledgement is removed.

P29, L3. Correct "Eppley".

Eppley is corrected.

P33, Fig. 10. Add the units for O2 minimum (umol/l).

The units was added to the legend of the figure.

---

## Author Comment (AC2) · 14 Jun 2019

Dear Referee,

Please find our comments/responses in blue throughout your text.
We have also attached a new release of the article.
This is a major revision of the initial article: the introduction has been completely revised, the sections are modified, a discussion section has been added. A synthesis of the results has been added by using a table and a Taylor diagram.
All your comments have been taken into account in the new release of the article.
If required we can also provide a version with the word track change, but I don't think that's helpful to you.
Thank you for your useful comments.

Elodie Gutknecht on behalf of co-authors.

The main objective of the article "A CMEMS forecasting system for the marine ecosystem of IBI European waters" is to demonstrate the performances of the CMEMS operational 1/36° coupled NEMO/PISCES of the IBI (extended) area from a 2010-2016, 7 years model simulation. The authors state that the model is in relatively good agreement with observations (for the main biogeochemical variables) and reproduce the large scale main biogeochemical processes, with a focus on seasonal cycles. In addition, an interesting oxygen deficiency indicator is presented. The article includes a wide variety of comparisons (15 figures) computed from an impressive set of observation. The document is well written and a very valuable effort has been done to make it as clear as possible. It results in a complete review of PISCES ability to simulate large scale main biogeochemical features in the IBI domain. In my opinion, despite these undeniable qualities, the article misses some important points and there are several major issues that should be (at least mostly) addressed before publication. The "general recommendations section" describes major points that, in my opinion, need to be improved before publication. The second section only provides some specific comments that could help to address the general recommendation needs. I did not go into very specific details as I think some materials should be added (and modified) before going further.

**General recommendations:**

A) In the actual form, the article is rather a review (interesting) of results from a 7 years PISCES simulation in the IBI region. It not clear to me whether the objectives are 1) demonstrate the ability of this system for regional forecast applications or 2) to assess PISCES ability to represent the main biogeochemical features of the IBI region => suitable for operational applications. In any case, important modifications need to be made and the introduction should be extended to better mention the objectives and the means used to address these objectives. From the sentence p2, line27, I guess the main objective is more on point 2) but the way it is written gives the impression that the authors want to assess the relevancy of this configuration for operational 7 day forecasts which is clearly not at all demonstrated here (ex 4.1.1. first sentence "Predicted sea surface..." should be rather something like "the model sea surface chlorophyll...").

**Author answer:**
The IBI36 system has been developed to monitor and forecast the ocean dynamics and marine ecosystems of the European waters. But before considering operational applications, a pre-operational qualification simulation is evaluated to assess PISCES ability to represent the main biogeochemical features of the IBI region. The purpose of this paper is to evaluate this qualification simulation.
The title and the Introduction of the submitted version are confusing. So to avoid any misunderstanding, we propose to modify the title of the paper by: "Modelling the marine ecosystem of IBI European waters for CMEMS operational applications". We also modified the abstract and the Introduction to make clearer the objectives of the paper.

This comment also underlines a lack of focus. The result section is more a presentation of various diagnosis than a scientific analysis of the results which might be induced by too unclear objectives of the paper. I understand that the paper is more an analysis of "consistency" but the reader can sometimes be irritated when a figure is nearly not discussed. If you decide to mainly make this scientific analysis in the discussion part i would suggest to dedicate a full section to this discussion (instead of discussion and conclusion) and to clearly state this point in introduction. Finally, taking into account a focus on objective 2), we sometimes lose the purpose of the paper and i am not sure that the article really answer whether or not this PISCES simulation is well adapted to operational simulation in the IBI area (i am sure it is). In my opinion this it mainly the consequence of an efficient but also too straight way of writing with a lack of methodology: "the role of this figure is to show this point, which demonstrates this information, which is related to my main objectives in this way". Nevertheless, i am sure that this could be corrected quite easily as there is also a very robust amount of information.

**Author answer:**
The entire text has been revised in order to clarify the objectives of the paper. The objective of each figure, each section and sub-section is now better apprehended. Figures are now more discussed. And a full section is now dedicated to the discussion, and it is clarified in the Introduction.

B) One major issue is the total absence of informations regarding uncertainties (except in figure 4). In my opinion, this point is crucial to assess the performance of any simulation even if the authors decide to only focus on point 2). As this might be difficult (and not necessary) to consider uncertainty for every comparison, I suggest to add one section dedicated to uncertainties. It should both discuss uncertainties of the PISCES simulation and the observation products. One simple question that should be addressed: is the model simulation included within the range of observation uncertainty? For instance, first order uncertainties could be deduced from the statistical level of dispersion (in a specific radius representative of error correlations). This particular suggestion might probably be better adapted for comparison with ocean colour data (or when a large amount of data is available and could be presented with histograms).

**Author answer:**
The issue of uncertainty is a truly complex one. The model uncertainties can be supported by the use of ensemble simulations. Here, the IBI36 system is a deterministic model. Only one trajectory is described. The uncertainties of the data are also complex to access, and are not always accessible.
In order to complete the skill assessment and better apprehend the uncertainty of the model, the bias and RMSE with to the ESA OC-CCI is added for Chl-a evaluation. Uncertainties (in terms of bias and RMSE) are the greatest in the coastal areas, which also correspond to the areas where the uncertainty of satellite measurements is highest (100% uncertainty compared to 30% for the open ocean; Moore et al., 2009). For NPP, the standard deviation between the three NPP products (VGPM, Eppley-VGPM and CbPM) is compared to the bias between the model and the mean of the three NPP products. This comparison shows that the modelled NPP is included within the range of observation uncertainty (as standard deviation between the three NPP products is considerable). Uncertainty is discussed in the discussion section (Sect. 5).

The suggested analysis deduced from error correlations is a work in progress in the framework of the development of an assimilation system for ocean colour data and in-situ data. This analysis represents a significant work that will be described in detail when setting up the future system with assimilation of biogeochemical tracers.

C) Particularly using a 1/36° model resolution, it is quite a shame not to show simple weekly or monthly mean maps (at least for chlorophyll) highlighting this PISCES simulation ability to catch the variability existing in ocean colour data. This would be relevant to insist on the benefits of using a 1/36° resolution. Are there such 1/36 PISCES simulation elsewhere? What are the benefits compare to lower resolution models? Why do you need this resolution here? I really think that it would be of great importance to compare your results with other existing configurations or models. This would also be a great help to demonstrate that your configuration is well adapted to simulate the IBI biogeochemical features (to solve some of the general recommendation A) ). If it is really not possible to perform comparisons this have to be clearly stated in introduction. More generally, the introduction completely misses the state of the art part. This definitely has to be corrected.

**Author answer:**

The PISCES model is intended to be used for both regional and global configurations at high or low spatial resolutions as well as for short-term (seasonal, interannual) and long-term (climate change, paleoceanography) analyses (Aumont et al., 2015). But to our knowledge and that of the PISCES developers, the PISCES model has never been used at such a resolution before.

Within the framework of CMEMS, three other MFC share a part of their domain with IBI:
- GLO-MFC which covers the world's oceans at 1/4° resolution and is also using the PISCES biogeochemical model,
- MED-MFC which covers the Mediterranean Sea at 1/24° with the Biogeochemical Flux Model (BFM; Vichi et al., 2007a,b),
- NWS-MFC which covers the North-West European Shelf at 1/15° latitudinal resolution and 1/9° longitudinal resolution (~ 7km) with the ERSEM ecosystem model (Baretta et al., 1995).

For the physical component, two intercomparison papers have been submitted in ocean Science - Special issue "The Copernicus Marine Environment Monitoring Service (CMEMS): scientific advances". They are Lorente et al. (2019) and Mason et al. (2019).

For the biogeochemical component, the comparison of the different model applications is a work in progress and will be the subject of a separate paper, including the contribution of the regional in relation to the global, as well as the comparison of three distinct biogeochemical models.

The Introduction has been revised and now includes a state of the art part.

**References:**

Baretta, J. W., W. Ebenhoh, et al. (1995). "The European regional seas ecosystem model, a complex marine ecosystem model." Netherlands Journal of Sea Research 33(3-4): 233-246.
Lorente, P., García-Sotillo, M., Amo-Baladrón, A., Aznar, R., Levier, B., Sánchez-Garrido, J. C., Sammartino, S., De Pascual, Á., Reffray, G., Toledano, C., and Álvarez-Fanjul, E.: Skill assessment of global, regional and coastal circulation forecast models: evaluating the benefits of dynamical downscaling in IBI surface waters, Ocean Sci. Discuss., https://doi.org/10.5194/os-2018-168, in review, 2019.
Mason, E., Ruiz, S., Bourdalle-Badie, R., Reffray, G., Garcia-Sotillo, M., and Pascual, A.: Copernicus (CMEMS) operational model intercomparison in the western Mediterranean Sea: Insights from an eddy tracker, Ocean Sci. Discuss., https://doi.org/10.5194/os-2018-169, in review, 2019.
Vichi, M., Pinardi, N., and Masina, S.: A generalized model of pelagic biogeochemistry for the global ocean ecosystem. Part I: theory, J. Marine Syst., 64, 89-109, https://doi.org/10.1016/j.jmarsys.2006.03.006, 2007a.
Vichi, M., Masina, S., and Navarra, A.: A generalized model of pelagic biogeochemistry for the global ocean ecosystem. Part II: numerical simulations, J. Marine Syst., 64, 110-134, https://doi.org/10.1016/j.jmarsys.2006.03.014, 2007b.

D) In order to make the document easier to read (in particular considering the probable adding figures (from my previous comments), i also suggest to add an annex section. Indeed some figures are only quickly and partially discussed (for instance : first paragraph of section 4,2,1 only 8 lines to described 3 figures ; in section 4,1,1, 4 figures are covered ; fig 7,8,9 f) are not discussed... ). This would also allow to better focus the article on its objectives. (This comment is connected with the general recommendation 1) ).

**Author answer:**
The objectives are now clearly presented in the Introduction, and figures are better discussed. We are aware that it is a substantial paper, with many figures. But it's necessary and voluntary. This paper represents the first validation of the biogeochemical component of the IBI36 system; it is intended to be complete and detailed, as it will serve as a basis for further studies.

**Main Specific comments:**
Considering the major recommendations, i here only specify, by section, the main specific comments as there will probably have a second review process. Generally, i do think that some accuracy is needed.

1) Introduction - See section major recommendations.
It has to be extended in order to clearly explain the objectives of the article and present a clear state of the art (in terms of existing forecast studies with PISCES and on models on the IBI region) in the area.

**Author answer:**
Introduction has been fully revised to clearly explain the objectives of the article. A state of the art in terms of existing forecast studies with PISCES and on models on the IBI region is now available.

2) IBI European waters
Interesting and quite complete part. It is a nevertheless a shame that the link between this part and the result section are nearly non-existent.

**Author answer:**
IBI European waters Section has been improved. It now introduce the boxes used for the evaluation of Chl-a and NPP. The link between this part and the result section in now improved.

3) The IBI36 configuration
- What is the influence of a 1/4° (bio) and 1/12° (physics) initialisation in the 1/36° simulation ? Do you have an idea how long this information is kept in the system, about the differences ?

Don't you think it can strongly impact the first timing and intensity of the blooms (especially with an initialization in January ? Why don't you make a few years spin-off ? - I would be very interested to have some informations on differences between a 1/4° and a 1/36° simulations. It would also help to justify the use of your configuration.

**Author answer:**
Initialization and open boundary conditions of the IBI36 system answer the CMEMS requirements. The physics comes from the CMEMS global physical component and the biogeochemical conditions come the CMEMS global biogeochemical system, this latter being also forced by the CMEMS global physical component.
The influence of a 1/4° (bio) and 1/12° (physics) initialisation and forcing in the 1/36° simulation is an interesting question. Impact analyses have been performed in the framework of the AMICO project, but only for short term simulations (1-2 years), and so the impacts were limited to the borders of the domain, but longer simulations would be necessary.
The date of the initialisation has been tested. The initialisation in January benefits from a low productivity. The seasonal dynamics triggered by seasonal warming and stratification has not yet begun. And so, the system is slowly being set up.
A few years spin-off are not performed here. The IBI36 system starts with the outputs of the global physical and biogeochemical systems, based on the same NEMO-PISCES model, and which has already turned 3 years. The main strengths and weaknesses of the model are found in both global and regional systems, such as the timing of the North Atlantic spring bloom. Systematic comparison between the global and regional systems is being performed, but we think that showing such metrics does not enter in the topic of this paper.

Some additional information about the nutrient forcing files would be welcome as i suppose that it could mainly explain most of the coastal deviations with observations. Do you have an idea of probable impacts of using 1/2_ data to 1/36_ grid, how do you deal with this ?

**Author answer:**

Nutrient inputs come from the annual climatology at ½° spatial resolution Global News 2. They are reported to the 1/36° grid at the river plumes considered in the IBI36 system (Rhone River and the German Bight) and along the coastline for all other inputs. Thus, we have endeavoured to conserve the nutrient flows between the original Global News 2 grid and the IBI36 grid. Mayorga et al. (2010) report that Global NEWS 2 underestimates nutrient runoffs in the Western Europe. Indeed, the only contribution of Global NEWS 2 is not sufficient to support the high coastal biological production of the IBI European waters. In order to reproduce the maximum Chl-a observed along the European coasts, additional inputs of nitrates and phosphates are provided into the IBI36 system at source points of the 33 main rivers and are linked to the physical flow. Location of the rivers can be found in Maraldi et al. (2013). They come from rivers monitored and listed by the European Environment Agency (www.eea.europa.eu) on the basis of annual averages.

Nutrient forcing description was not clear and maybe confusing. The description is now improved in Section 3.2. Nutrient inputs from rivers are of primary importance for coastal ecosystems. Unfortunately, they are often introduced in a too simplistic way in regional models due to a crucial lack of available measurements. This is a weakness that really needs to be considered and improved, but that is not intrinsic to the IBI36 system.

4) IBI36 evaluation
4.1 Satellite estimations
- How is the temporal correlation figure 2d calculated ? From monthly averages ? - On which grid are the differences calculated (model or verification) ? Are there impacts resulting from this grid changing ? In particularly for Net primary production comparison (1/6 degree compared to 1/36 degree). Please clarify this point as non linearities can have significant effects. - It is difficult to see something in figure 3. The discussion on the bloom timing could also be done using different boxes of figure 4. This would permit to remove one figure (or in annex). - p6. lines 18-21 you say that Net primary production estimates are model products ? If it is true why do you include these data sets in a section 4,1 called satellite estimations ?

**Author answer:**

Temporal correlation is calculated from monthly averages between 2010 and 2016. ). It is now clearly stated at the beginning of Section 4.1.

We have chosen to interpolate the model on the data grid, i. e. 1 km for the comparison to ESA OC-CCI ocean colour product and 1/6° for NPP.

Yes, the discussion on the bloom timing could be done only using different boxes of Figure 4, and we could remove Figure 3. But Figures 3 and 4 are complementary. Figure 3 clearly shows the seasonal phase shift and the high interannual variability in the data while the model is more seasonal. This is more evidenced than in Figure 4.

Net primary production estimates are model products because an algorithm (or model) is used to determine NPP from ocean colour data. But it is considered primarily as a product derived from satellite estimates. To avoid any misunderstanding, the term "NPP product" instead of NPP model" is used in the revised version.

4,2 In situ historical data
- In fig 8, for oxygen, it would be clearer if you could modify the colorbar. At a first view we think that the data and the model are very different while the bias is only 4% - It is a shame you do not discuss at all some possible reasons why the model does not catch the low oxygen period in 2014-2015. Especially when you thereafter discuss about oxygen deficiencies: : : -Don 't you think that one of the main limitation comes from the nutrient forcing files ? Could you specify a

little bit more (than it is in 3.2) as it seems to be quite important ? Where are the anthropic inputs located in the model ? What is the impact of these additional anthropic inputs ? It could be relevant to go a little bit deeper into this point.

**Author answer:**
The colorbar of Figure 9 has been modified.

The model does not catch the lower oxygen concentrations observed in ICES in 2014-2015. It can come from the Baltic Sea or from river inflows. But I didn't find any reference to this event in the literature.

The oxygen-deficient areas are quite well estimated by the model in terms of spatial distribution and extension. They are located mainly along the coastline and are certainly impacted by river inflows but also sedimentary processes. More realistic external forcing files would obviously improve the estimation of vulnerable areas. But as explained above, they are introduced in a too simplistic way due to a crucial lack of available measurements.

Nutrient forcing description was not clear and maybe confusing. The terms "natural inputs" and "anthropogenic inputs" are now removed because they are not really accurate and could be confusing. The description is now improved in Section 3.2.

This adaptation was necessary to simulate higher coastal Chl-a and reproduce the maximum Chl-a observed along the European coasts. But it is not totally satisfactory because a strong seasonal signal seems to emerge from ICES comparisons. The future system will improve this external forcing.

4,3 Argo data
How are the Argo data co-located with the model data ? Do you grid Argo data on the model grid ? Could you re-precise dates ? Although correlations are still high, results are much less good than previously. Do you have an idea why ? Small scale features ? Have you compared the Argo data with some of the previous observations ? (it also could help in defining uncertainty levels).

**Author answer:**
As for the satellite comparison, we have chosen to interpolate the model on the BGC-Argo data grid. We used daily averaged model outputs at the nearest model grid point, and a linear interpolation for the vertical.

BGC-Argo float comparisons open new perspectives. Bu the comparison should be considered with some cautions because the product quality procedures are on-going work. They are not fully established or homogenized for all floats. Temporal drifts, constant or even non-constant vertical bias, and negative concentrations (see the nitrate in Fig. 15d) are still observed in the BGC-Argo data.

4,4 Discussion and conclusion

- I would suggest to separate the discussion and the conclusion since the authors have clearly decided not to deeply analyze the results in the result section. The discussion proposed here is interesting but should be extended and also make a clear link better with the objectives that should be first clarified in introduction.

**Author answer:**
Discussion and Conclusions are now separated, and the link with the objectives of the paper as clarified in introduction is now improved.

We decided to only describe the results in Section 4, and then deeply analyse the results in the Discussion section. This structure seems simpler because some aspects come up several times in the evaluation (smoothed vertical profiles, rivers, sediments…). So we discuss them once and for all in the dedicated section.

---

## Referee Report (RR1)

**Comments on the reviewed manuscript "Modelling the marine ecosystem of IBI European waters for CMEMS operational applications" by E. Gutknecht, G. Reffray, A. Mignot, T. Dabrowski, and M.G. Sotillo.**

The reviewed version of the manuscript is significantly improved, in particular for the clarification of the focus, now specifically targeted to the pre-operational assessment of the model system. I encourage the authors to also provide, in a next contribution, a quality assessment of the NRT products, which may have an important impact for CMEMS users.
In any case, I think this work will remain as a reference for biogeochemical modelling in the IBI area.

In the list of minor comments, I suggested some possible changes to the text, in particular concerning the use of the very short sentences, that may reduce the readability. A slightly major modification would be to reduce most of the Conclusions (lines 570 to 602) to 4/5 main results in form of items, to be considered as synthetical "take-home messages" – but I leave this option to authors.
Further, please pay particular attention to the three following comments, the second being potentially more effort-demanding:

1. Table 1 refers to the global synthesis performance rather than to the 12 specific areas: I remain of the idea that expressing the quality indicators (mean and std, mean error, RMSE, %Bias and Correlation) for each of the 12 boxes would be much more interesting – maybe including at least one more indicator (RMSE) in Figs. 4 and 6?
2. P14-L567-569: "This paper represents the first validation of the biogeochemical component of the IBI36 system**: the** objective is to show that PISCES can be used for operational applications, and that there is no contraindication to using the PISCES model at such a resolution." Further, why do you use "contraindication"? is there any specific reason? You already wrote in Introduction that "While PISCES has so far been used to answer a wide range of scientific questions, it has never before been used at such a resolution." but I cannot see any point to refer its use as a "contraindication"…please reformulate. Finally, and this may be worth of a separate discussion: you showed the assessment (in terms of consistency) of a 1/36 resolution model, but in the manuscript the advantages and the benefits of using such a high resolution are poorly underlined and discussed: how this version compares with the GLO product in the same area, for example? Would it be possible for the authors to briefly discuss more about the added value of a higher resolution system? I think the reader would be interested to see some consideration about this issue.
3. Captions of Fig. 12 and 15 report "r" as the Pearson correlation coefficient, but I cannot see any reference to r in the plots.

As a marginal note: for a better readability of the response letter, I would have preferred to find a more precise correspondence between the issues raised in the first review and the corrections done in the reviewed version (for example reporting the new version for each point, highlighting the number of pages/lines where modifications have been performed), since it has been a little bit cumbersome to check all the updates. However, the track changes version was helpful enough.

Minor comments:

4. P1-L12: better "," than ";"...: "...European waters, and more specifically..."

5. P1-L17: the use of "skill" would be more appropriate to evaluate model performance for operational systems (i.e. forecast skill, see for example Lorente et al., 2016[1]), rather than for the pre-operational assessment. I suggest the authors consider to use "consistency and skill", as already done in other parts of the manuscript.

6. P2-L47: maybe "...in Fig. 1a, and further..." ?

7. P2-L65: maybe one space less: "(the INDESO project; Gutknecht et al., 2016)."

8. P2-L67: possibly remove "before" or put it at the end of the sentence

9. P2-L73: Also Lazzari et al., 2012 could be referred here: Lazzari, P., Solidoro, C., Ibello, V., Salon, S., Teruzzi, A., Béranger, K., Colella, S., and Crise, A.: Seasonal and interannual variability of plankton chlorophyll and primary production in the Mediterranean Sea: a modelling approach, Biogeosciences, 9, 217–233, https://doi.org/10.5194/bg-9-217-2012, 2012.

10. P3-L88: "synthetic" ?

11. P3-L97: possibly unify the 2 sentences: "...4.1), and are named..."

12. P3-L100: remove the comma between "and" and "when": "…starts in spring, when seasonal re-stratification begins and when the Mixed Layer Depth (MLD)…"

13. P3 and P4 - L104, 114, 121, 125, 129, 141, 142: Fig. 1**a**

14. P4-L128: add year to reference "(UNEP LME report, 2008)"

15. P4-L132: "one biomass peak", please consider to change to "a biomass peak" or, if pertinent, "a major/main/predominant biomass peak"

16. P5-L168-171: the 3 sentences could be unified for better readability, as an example (but please consider also other possibilities): "There are five nutrients that limit phytoplankton growth (nitrate and ammonium, phosphate, silicate and iron)**, and the** model distinguishes two phytoplankton size compartments (nanophytoplankton and diatoms) expressed in carbon, iron, Chl-a and silicon content (the latter only for diatoms) and two zooplankton size classes (microzooplankton and mesozooplankton)**; the** bacterial pool is not explicitly modelled."

17. P5-L204: decide if use "NEWS" or "News".

18. P6-L227: "…(Gohin et al., 2008)."

19. P6-L241: "validate**s**"

20. P6-L242-252: these sentences appear too short and, to improve readability I suggest to slightly re-write this part
* * *
[1] https://www.tandfonline.com/doi/full/10.1080/1755876X.2016.1215224

21. P7-L264: "Finally, corrections are applied on each variable to correct from calibration biases and sensor drifts." Repetition: possible suggestion: "Finally, corrections are applied on each variable to **remove/reduce** calibration biases and sensor drifts."

22. P7-L265-268: these sentences appear too short and, to improve readability I suggest to slightly re-write this part

23. P7-L272: "**d**aily"

24. P7-L286: Fig. 1**a**

25. P8-L301-306: I would mention also the biases around Faroe Islands: do you have any explanation for that?

26. P8-L321-324: suggested changes: "Coastal ecosystems of the Bay of Biscay (box 7) **show** a peak biomass during spring bloom, **while** the upwelling off Portugal and Morocco (boxes 8 and 10) present**s** a maximum in spring with more interannual variability off Morocco. **In the Gulf of Cadiz (box 9) and the Western Mediterranean (boxes 11 and 12), IBI36** succeed**s** in reproducing the seasonal cycle of Chl-a (Fig. 4), with a high correlation coefficient (r > 0.71) **with** the satellite product."

27. P8-325-326: "In the open North Sea (**box 4**), the first peak is usually reproduced, but the data present a strong interannual variability. In the southern North Sea (**box 5**)…"

28. P10-L377-378: "In addition, the model does not capture the lower sea surface oxygen concentrations  measured during 2014-2015 period (Fig. 10)."

29. P10-L399-400: "The statistics are low (Fig. 7) while the density plot, surface distribution and time series (Fig. 8 to 10) are positive." …not clear: can you formulate better?

30. P10-L406-407: suggestion… "Oxygen content is a key element in biogeochemical cycles and can be an indicator of the health of marine ecosystems: **for this reason we analyse** the minimum oxygen concentrations."

31. P11-L428: "illustrate**s**"

32. P11-L428: I would say "**Model** oxygen and nutrients show…"

33. P11-L429: "0.95, **see** Fig. 7." Or simply "0.95 (Fig. 7)."

34. P12-L458: I would avoid adverbs as "now", "comparison **with**…"

35. P12-L482: "…develops **in** summer in the model simulation and is not present in the observations"

36. P13-L492-494: suggestion "…providing the trails for improvement to be explored**, which are here** discussed."

37. P13-L517: should it be "**even** out of phase" ?

38. P13-L520: "performs"

39. P13-L522-526: again, to improve readability I suggest: "This behaviour is also observed in the global model **at** ¼° (Perruche et al., 2016) used **for/to set-up** the initial and open boundary conditions**, and** can originate from the physical or biogeochemical models. Different approaches are currently under study**: in particular, vertical** diffusion could explain the loss of peaks and minima in vertical profiles, but biogeochemical processes (**e.g.,** parameterization of remineralisation processes, rate of sinking of particulate detritus, vertical migration of zooplankton which export organic matter at depth) are* also investigated." *using "are" it means you are showing this somewhere: if this is not the case, I would use "will be" or something similar.

40. P14-L538: "This assumption may be too restrictive**: as** an alternative ..."

41. P14-L566: I think "description" or similar may be better than "vision"…

42. P14-L570-571: again, some suggested improvement for readability "Chl-a and NPP are compared to satellite estimates, **describing here their** mean spatial distribution and seasonal cycle."

43. P14-L574-576: "Some of these areas are outside of the IBI Service Domain (that is the geographical domain covered by the CMEMS IBI-MFC products)**, but** in order to take advantage of their *in situ* observational coverage**, we also evaluated** the IBI Extended Domain."

44. P15-L579: "MOC"? maybe you refer to Deep Oxygen Maximum (DOM)?

45. P15-L583-586: ". The **model** averaged spatial distribution is close to CbPM product, but spatial distribution, cross-shore gradients and the seasonal variations are better correlated to the VGPM-based products. The modelled NPP is thus within the range of variability of the satellite derived estimates." There is a repetition between first and last sentences: I would remove the first sentence and add "model" in the following.

46. P15-L588: highlight**s**

47. Caption Fig.1, L905: "(boxes 11 and 12**)**."; L908: why "font" ?... maybe "blue colour shading" or something similar?

---

## Author Response (AR2)

**Comments on the reviewed manuscript "Modelling the marine ecosystem of IBI European waters for CMEMS operational applications" by E. Gutknecht, G. Reffray, A. Mignot, T. Dabrowski, and M.G. Sotillo.**

Dear Referee,

Please find our comments/responses in blue throughout your text.
We have also attached a new release of the article.
All your comments have been taken into account in the new release of the article.
Thank you for your useful comments.

Elodie Gutknecht on behalf of co-authors.

The reviewed version of the manuscript is significantly improved, in particular for the clarification of the focus, now specifically targeted to the pre-operational assessment of the model system. I encourage the authors to also provide, in a next contribution, a quality assessment of the NRT products, which may have an important impact for CMEMS users. In any case, I think this work will remain as a reference for biogeochemical modelling in the IBI area.

In the list of minor comments, I suggested some possible changes to the text, in particular concerning the use of the very short sentences, that may reduce the readability. A slightly major modification would be to reduce most of the Conclusions (lines 570 to 602) to 4/5 main results in form of items, to be considered as synthetical "take-home messages" – but I leave this option to authors.

**Author answer:**
- As detailed below (in the minor comments), I reformulated the very short sentences.
- I tried to highlight the 4 main results in the form of items.

Further, please pay particular attention to the three following comments, the second being potentially more effort-demanding:

1. Table 1 refers to the global synthesis performance rather than to the 12 specific areas: I remain of the idea that expressing the quality indicators (mean and std, mean error, RMSE, %Bias and Correlation) for each of the 12 boxes would be much more interesting – maybe including at least one more indicator (RMSE) in Figs. 4 and 6?

**Author answer:**
The RMSE in addition to the correlation is now indicated n Figs. 4 and 6.

2. P14-L567-569: "This paper represents the first validation of the biogeochemical component of the IBI36 system**: the** objective is to show that PISCES can be used for operational applications, and that there is no contraindication to using the PISCES model at such a resolution."

> **Author answer:**
> The two sentences are now combined in one sentence by the use of ": **the**".
> P15 – L966-967 of the new version.

Further, why do you use "contraindication"? is there any specific reason? You already wrote in Introduction that "While PISCES has so far been used to answer a wide range of scientific questions, it has never before been used at such a resolution." but I cannot see any point to refer its use as a "contraindication"...please reformulate.

> **Author answer:**
> We reformulated "the objective is to show that PISCES can be used for operational applications, and that there is no contraindication to using the PISCES model at such a resolution" by "…, and that it is a suitable tool at such a resolution".
> P15 – L966-967 of the new version.

Finally, and this may be worth of a separate discussion: you showed the assessment (in terms of consistency) of a 1/36 resolution model, but in the manuscript the advantages and the benefits of using such a high resolution are poorly underlined and discussed: how this version compares with the GLO product in the same area, for example? Would it be possible for the authors to briefly discuss more about the added value of a higher resolution system? I think the reader would be interested to see some consideration about this issue.

> **Author answer:**
> The advantages and the benefits of using such a high resolution are now discussed in the Introduction.
> P2 – L71-80 of the new version.

3. Captions of Fig. 12 and 15 report "r" as the Pearson correlation coefficient, but I cannot see any reference to r in the plots.

> **Author answer:**
> Sorry, it was an oversight. The legends of Fig. 12 and 15 are now corrected.

As a marginal note: for a better readability of the response letter, I would have preferred to find a more precise correspondence between the issues raised in the first review and the corrections

done in the reviewed version (for example reporting the new version for each point, highlighting the number of pages/lines where modifications have been performed), since it has been a little bit cumbersome to check all the updates. However, the track changes version was helpful enough.

**Author answer:**
The changes were so important that they were difficult to give a precise correspondence between the first version and the corrections.
But I apologize to you; and I take your remark into account.

**Minor comments:**

4. P1-L12: better "," than ";"...: "...European waters, and more specifically..."
";" is changed to ",".

5. P1-L17: the use of "skill" would be more appropriate to evaluate model performance for operational systems (i.e. forecast skill, see for example Lorente et al., 20161), rather than for the pre-operational assessment. I suggest the authors consider to use "consistency and skill", as already done in other parts of the manuscript.
"skill performances" is changed to "consistency and skill assessment"

6. P2-L47: maybe "...in Fig. 1a, and further..." ?
A "," is added.

7. P2-L65: maybe one space less: "(the INDESO project; Gutknecht et al., 2016)."
The space is removed.

8. P2-L67: possibly remove "before" or put it at the end of the sentence
"before" is now put at the end of the sentence.

9. P2-L73: Also Lazzari et al., 2012 could be referred here: Lazzari, P., Solidoro, C., Ibello, V., Salon, S., Teruzzi, A., Béranger, K., Colella, S., and Crise, A.: Seasonal and interannual variability of plankton chlorophyll and primary production in the Mediterranean Sea: a modelling approach, Biogeosciences, 9, 217–233, https://doi.org/10.5194/bg-9-217-2012, 2012.
The reference ot Lazzari et al. (2012) is added.

10. P3-L88: "synthetic" ?
 "a synthesis overview" is changed to "an overview".

11. P3-L97: possibly unify the 2 sentences: "...4.1), and are named..."
The 2 sentences are unified.

12. P3-L100: remove the comma between "and" and "when": "…starts in spring, when seasonal re-stratification begins and when the Mixed Layer Depth (MLD)…"
The comma is removed.

13. P3 and P4 - L104, 114, 121, 125, 129, 141, 142: Fig. 1**a**
We now refer to Fig. 1a.

14. P4-L128: add year to reference "(UNEP LME report, 2008)"
The year is added.

15. P4-L132: "one biomass peak", please consider to change to "a biomass peak" or, if pertinent, "a major/main/predominant biomass peak"
"one biomass peak" is changed to "a major biomass peak".

16. P5-L168-171: the 3 sentences could be unified for better readability, as an example (but please consider also other possibilities): "There are five nutrients that limit phytoplankton growth (nitrate and ammonium, phosphate, silicate and iron)**, and the** model distinguishes two phytoplankton size compartments (nanophytoplankton and diatoms) expressed in carbon, iron, Chl-a and silicon content (the latter only for diatoms) and two zooplankton size classes (microzooplankton and mesozooplankton)**; the** bacterial pool is not explicitly modelled."
The 3 sentences are now combined in: "The model considers five nutrients that limit phytoplankton growth (nitrate, ammonium, phosphate, silicate and iron) and four living compartments: two phytoplankton size classes (nanophytoplankton and diatoms) and two zooplankton size classes (microzooplankton and mesozooplankton); the bacterial pool is not explicitly modelled."

17. P5-L204: decide if use "NEWS" or "News".
We changed to "Global NEWS 2".

18. P6-L227: "…(Gohin et al., 2008)."
I removed the ".".

19. P6-L241: "validate**s**"
It is now corrected.

20. P6-L242-252: these sentences appear too short and, to improve readability I suggest to slightly re-write this part
"EMODnet Chemistry has adopted and adapted SeaDataNet standards and services. Regional aggregated datasets are available for the North-East Atlantic and the Mediterranean (www.emodnet-chemistry.eu/products). The aggregated datasets are regional *in situ* observation collections receiving additional quality control of metadata and data. The North-East Atlantic Ocean regional dataset is aggregated, standardized and quality controlled by IFREMER / IDM / SISMER - Scientific Information Systems for the SEA (2018) from France, and the Mediterranean Sea dataset by Hellenic Centre for Marine Research, Hellenic National

Oceanographic Data Centre (HCMR/HNODC) (2018) from Greece. The regional datasets contain oxygen, nitrate, phosphate, silicate and ammonium profiles all in µmol l-1 and Chl-a profiles in mg Chl m-3. The North-East Atlantic dataset includes data from the OVIDE section between Portugal and Greenland in spring 2010 and the PELGAS cruises each spring on the Bay of Biscay. For PELGAS comparison, Chl-a data come from Niskin bottles. For the Mediterranean Sea regional dataset, only Chl-a is presented as it has the best spatial cover as compared to other variables."

is changed to:

"The Chemistry component of EMODnet has adopted and adapted SeaDataNet standards and services, and delivers regional aggregated datasets receiving additional quality control of metadata and data (www.emodnet-chemistry.eu/products). These *in situ* observation collections contain oxygen, nitrate, phosphate, silicate and ammonium profiles all in µmol l$^{-1}$ and Chl-a profiles in mg Chl m$^{-3}$. The North-East Atlantic Ocean dataset includes data from the OVIDE section between Portugal and Greenland in spring 2010 and from the springtime PELGAS cruises on the Bay of Biscay. For the Mediterranean Sea dataset, only Chl-a is presented as it has the best spatial cover as compared to other variables."

"The North-East Atlantic Ocean regional dataset is aggregated, standardized and quality controlled by IFREMER / IDM / SISMER - Scientific Information Systems for the SEA (2018) from France, and the Mediterranean Sea dataset by Hellenic Centre for Marine Research, Hellenic National Oceanographic Data Centre (HCMR/HNODC) (2018) from Greece." is moved to Data availability Section.

21. P7-L264: "Finally, corrections are applied on each variable to correct from calibration biases and sensor drifts." Repetition: possible suggestion: "Finally, corrections are applied on each variable to **remove/reduce** calibration biases and sensor drifts."
"Finally, corrections are applied on each variable to correct from calibration biases and sensor drifts." Is changed to ""Finally, corrections are applied on each variable to reduce calibration biases and sensor drifts."".

22. P7-L265-268: these sentences appear too short and, to improve readability I suggest to slightly re-write this part
"The APEX float observations are adjusted following Johnson et al. (2017). The three variables are "delayed mode" data. For the 265 PROVOR float observations, oxygen and nitrate are "Real time" data and Chl-a is "adjusted" data. They are adjusted following Mignot et al. (2018). The first five months of nitrate measurement by the PROVOR float were masked due to spurious values." Is changed to:
"For the APEX float observations, the three variables are "delayed mode" data, and are adjusted following Johnson et al. (2017). For the PROVOR float observations, oxygen and nitrate are "Real time" data and Chl-a is "adjusted" data, they are adjusted following Mignot et al. (2018), and the first five months of nitrate measurement were masked due to spurious values."

23. P7-L272: "**d**aily"

The capital letter is removed.

24. P7-L286: Fig. 1**a**
We now refer to Fig. 1a.

25. P8-L301-306: I would mention also the biases around Faroe Islands: do you have any explanation for that?
Biases around Faroe Islands are now mentioned. I have no explanation for that at that moment. A detailed analysis would be necessary.

26. P8-L321-324: suggested changes: "Coastal ecosystems of the Bay of Biscay (box 7) **show** a peak biomass during spring bloom, **while** the upwelling off Portugal and Morocco (boxes 8 and 10) present**s** a maximum in spring with more interannual variability off Morocco. **In the** Gulf of Cadiz (box 9) and the Western Mediterranean (boxes 11 and 12), **IBI36** succeed**s** in reproducing the seasonal cycle of Chl-a (Fig. 4), with a high correlation coefficient (r > 0.71) **with** the satellite product."
These changes are taken into account.

27. P8-325-326: "In the open North Sea (**box 4**), the first peak is usually reproduced, but the data present a strong interannual variability. In the southern North Sea (**box 5**)…"
The extra parenthesis is removed.

28. P10-L377-378: "In addition, the model does not capture the lower sea surface oxygen concentrations **than that** measured during 2014-2015 period (Fig. 10)."
We changed to :" In addition, the model does not capture the lower sea surface oxygen concentrations measured during 2014-2015 period (Fig. 10)."

29. P10-L399-400: "The statistics are low (Fig. 7) while the density plot, surface distribution and time series (Fig. 8 to 10) are positive." …not clear: can you formulate better?
The sentence is reformulated by : "The statistics are not satisfying (Fig. 7) while the density plot, surface distribution and time series (Fig. 8 to 10) give a quite positive evaluation."

30. P10-L406-407: suggestion… "Oxygen content is a key element in biogeochemical cycles and can be an indicator of the health of marine ecosystems: **for this reason we analyse** the minimum oxygen concentrations."
We changed to: "Oxygen content is a key element in biogeochemical cycles and can be an indicator of the health of marine ecosystems: for this reason the minimum oxygen concentrations are now analysed."

31. P11-L428: "illustrate**s**"
The correction is done.

32. P11-L428: I would say "**Model** oxygen and nutrients show…"
The modification is done.

33. P11-L429: "0.95, **see** Fig. 7." Or simply "0.95 (Fig. 7)."
I removed the comma.

34. P12-L458: I would avoid adverbs as "now", "comparison **with**…"
I simplified to: "The free-drifting BGC-Argo profiling floats allow continuous monitoring of dissolved oxygen, nitrate and Chl-a of the upper 1000 meters of the ocean."

35. P12-L482: "…develops **in** summer in the model simulation and is not present in the observations"
The correction is done.

36. P13-L492-494: suggestion "…providing the trails for improvement to be explored**, which are here** discussed."
The suggestion is taken into account.

37. P13-L517: should it be "**even** out of phase" ?
Yes, the correction is done.

38. P13-L520: "performs"
Yes, the correction is done.

39. P13-L522-526: again, to improve readability I suggest: "This behaviour is also observed in the global model **at** ¼° (Perruche et al., 2016) used **for/to set-up** the initial and open boundary conditions**, and** can originate from the physical or biogeochemical models. Different approaches are currently under study**: in particular, vertical** diffusion could explain the loss of peaks and minima in vertical profiles, but biogeochemical processes (**e.g.,** parameterization of remineralisation processes, rate of sinking of particulate detritus, vertical migration of zooplankton which export organic matter at depth) are* also investigated." *using "are" it means you are showing this somewhere: if this is not the case, I would use "will be" or something similar.
All these modifications are taken into account.

40. P14-L538: "This assumption may be too restrictive**: as** an alternative ..."
The modification is done.

41. P14-L566: I think "description" or similar may be better than "vision"…
I changed "vision" to "description".

42. P14-L570-571: again, some suggested improvement for readability "Chl-a and NPP are compared to satellite estimates, **describing here their** mean spatial distribution and seasonal cycle."
The modification is done.

43. P14-L574-576: "Some of these areas are outside of the IBI Service Domain (that is the geographical domain covered by the CMEMS IBI-MFC products)**, but** in order to take advantage of their *in situ* observational coverage**, we also evaluated** the IBI Extended Domain."
The modification is done.

44. P15-L579: "MOC"? maybe you refer to Deep Oxygen Maximum (DOM)?
Yes, I refer to DOM. I corrected the text.

45. P15-L583-586: "Simulated NPP lies between the three NPP products. The **model** averaged spatial distribution is close to CbPM product, but spatial distribution, cross-shore gradients and the seasonal variations are better correlated to the VGPM-based products. The modelled NPP is thus within the range of variability of the satellite derived estimates." There is a repetition between first and last sentences: I would remove the first sentence and add "model" in the following.
I removed the first sentence and added "model" in the following as you suggested.

46. P15-L588: highlight**s**
The modification is done.

47. Caption Fig.1, L905: "(boxes 11 and 12**)**."; L908: why "font" ?... maybe "blue colour shading" or something similar?
I added a parenthesis and changed "font" to "blue colour shading".

[revised manuscript text omitted]